



**Intercomparison of Terrestrial Carbon Fluxes and Carbon Use Efficiency Simulated by CMIP5 Earth System Models**

Dongmin Kim[1], Myong-In Lee[1*], Su-Jong Jeong[2], Jungho Im[1], Dong Hyun Cha[1] and Sanggyun Lee[1]

[1]School of Urban and Environmental Engineering, Ulsan National Institute of Science and Technology, Ulsan, Korea

[2]School of Environmental Science and Engineering, Southern University of Science and Technology, Nanshan, Shenzhen, Guangdong, China

-------------------------------------------------------------------------------------------------------------------------
Corresponding author: Dr. Myong-In Lee
School of Urban and Environmental Engineering
Ulsan National Institute of Science and Technology,
50 UNIST-gil, Ulsan 44919, Korea
Email: milee@unist.ac.kr



**Abstract**
This study compares historical simulations of the terrestrial carbon cycle produced by 10
Earth System Models (ESMs) that participated in the fifth phase of the Coupled Model
Intercomparison Project (CMIP5). Using MODIS satellite estimates, this study validates the
simulation of gross primary production (GPP), net primary production (NPP), and carbon use
efficiency (CUE), which depend on plant function types (PFTs). The models show noticeable
deficiencies compared to the MODIS data in the simulation of the spatial patterns of GPP and
NPP and large differences among the simulations, although the multi-model ensemble (MME)
mean provides a realistic global mean value and spatial distributions. The larger model spreads
in GPP and NPP compared to those of surface temperature and precipitation suggest that the
differences among simulations in terms of the terrestrial carbon cycle are largely due to
uncertainties in the parameterization of terrestrial carbon fluxes by vegetation. The models also
exhibit large spatial differences in their simulated CUE values and at locations where the
dominant PFT changes, primarily due to differences in the parameterizations. While the MME-
simulated CUE values show a strong dependence on surface temperatures, the observed CUE
values from MODIS show greater complexity, as well as non-linear sensitivity. This leads to
the overall underestimation of CUE using most of the PFTs incorporated into current ESMs.
The results of this comparison suggest that more careful and extensive validation is needed to
improve the terrestrial carbon cycle in terms of ecosystem-level processes.

Keywords: earth system models, carbon use efficiency, CMIP5, MODIS, GPP, NPP






### 1. Introduction

Earth system models (ESMs) have been developed in the past several decades to simulate vegetation changes in space and time through carbon cycle-related interactions between the biosphere and the atmosphere. The temporal variations in atmospheric $CO_2$ in the models are driven by $CO_2$ emissions from natural and anthropogenic sources, as well as uptake by vegetated land surfaces and the ocean. Net imbalances in carbon fluxes drive the secular trend in $CO_2$. The magnitude of the imbalance is model-dependent and results in differences in the future warming projected by various ESMs. Previous studies showed that the observed trend of atmospheric $CO_2$ was not reproduced correctly during the past century, given the historical record. There was also substantial spread among models, even though they were forced by identical anthropogenic emissions (Friedlingstein et al., 2006, 2014; Hoffman et al., 2013; Zhao and Zeng, 2014). The model bias persists into their future projections. Hoffman et al. (2013) pointed out that the spread of projected $CO_2$ concentrations among fifteen Coupled Model Intercomparison Project (CMIP5; Taylor et al., 2012) ESMs in 2100 was approximately 20 % of their multi-model average. Friedlingstein et al. (2014) showed that the degree of surface temperature warming by 2100 was different by more than a factor of two, depending on the models and representative concentration pathway (RCP) 8.5 scenarios used.

Previous studies (Friedlingstein et al., 2006; Booth et al., 2012; Hoffman et al., 2013; Anav et al., 2013; Aroa et al., 2013; Friedlingstein et al., 2014) have suggested that the uncertainty in $CO_2$ concentrations simulated by ESMs should be largely attributed to the terrestrial carbon uptake, rather than to the uptake by ocean. Hoffman et al. (2013) and Friedlingstein et al. (2014) compared the carbon uptake by land and ocean, simulated by ESMs and found that the amount of carbon accumulated by the ocean is positive in all models by 2100, whereas the models exhibited a large spread in the amount of carbon taken up by the land; the results even had





different signs. Aroa et al. (2013) indicated that the simulated sensitivity of terrestrial carbon
storage to the atmospheric $CO_2$ concentration was 3-4 times larger than that of ocean. This
suggests that the terrestrial carbon cycle is one of the important factors that need improvement
for minimizing uncertainty in future climate predictions.
It is generally recognized that changes in the carbon pools in the biosphere should play a key
role in determining atmospheric $CO_2$ concentration levels in the future. Shao et al. (2013)
showed that the net biome production (NBP) simulated by CMIP5 ESMs is enhanced in the
21st century and that the biomass particularly increases over tropical rainforests and vegetated
surfaces in the mid-latitudes through the $CO_2$ fertilization effect. Not only long-term increases
in biomass but also future changes in its seasonal cycle would significantly affect $CO_2$
concentrations. Zhao and Zeng (2014) indicated that the amplitude of the seasonal cycle of
atmospheric $CO_2$ tends to increase in the future, due to an increase of 68 % in the seasonal
cycle of NBP during the growing season in their future simulations. Comprehensive model
intercomparisons on the simulation of biome production at various ecosystem levels are needed
to explain the differences among simulations and minimize projection uncertainties.
The exchange of carbon between the atmosphere and terrestrial ecosystems consists of
complicated biogeochemical processes operating over a heterogeneous surface, and the quality
and the performance of the global model simulations is often diagnosed using carbon cycle
variables such as gross primary production (GPP) and autotrophic respiration (Ra) by plants.
Net primary production (NPP) is defined as GPP minus Ra. Heterotrophic respiration (Rh),
involving the decomposition of soil litter, is also an important process involved in the carbon
cycle. By validation using ground and satellite observational data, previous studies identified
the systematic biases of ESMs and discussed the possible reasons for these biases. Anav et al.
(2013) indicated that current ESMs tend to overestimate terrestrial biomass and global GPP



(Anav et al., 2013). Shao et al. (2013) showed that ESMs exhibit large disagreements in the
relationship between carbon cycle variables and hydrological variables, such as precipitation
and soil moisture, emphasizing the importance of the hydrological cycle in terms of its effects
on the terrestrial carbon cycle. The simulated soil carbon amount in the subsurface root zone,
which is the major source of plant growth, showed systematic biases and large model spread,
from 40 to 240 %, compared with observational data (Todd-Brown et al., 2013). That study
suggested that it might be responsible for the large spread of atmospheric CO2 concentrations
simulated by the models.
While most previous intercomparison studies involving ESMs have focused on the
validation of the global mean budget of terrestrial carbon pools and fluxes (Anav et al., 2013;
Shao et al., 2013; Todd-Brown et al., 2013), which is useful for evaluating the overall
performance of ESMs and quantifying simulation uncertainties, more detailed analyses
addressing regional scales and different vegetation types are needed to identify the key sources
of systematic biases in the models. Anav et al. (2013) evaluated regional changes in
biogeochemical variables for two hemispheres and the tropical region separately. In particular,
an investigation of systematic biases in different types of ecosystems is required to improve
the existing parameterizations of terrestrial carbon fluxes by vegetation. In contrast to the many
observational studies in biology that address various plant function type (PFT) levels (De Lucia
et al., 2007; Zhang et al., 2009; Zhang et al., 2014), studies that benchmark model simulations
of PFT levels have obtained less attention, and this is one of the primary motivations of this
study.
For a better elucidation of systematic biases in the models, this study focuses particularly on
the comparison of carbon use efficiency (CUE), which is sensitive to the various PFTs. For the
short-term carbon cycle, Ra is a primary measure of the release of carbon to the atmosphere,





and its magnitude is known to be about half of GPP for most vegetated surfaces (King et al.,
2006; Piao et al., 2010). CUE is defined as the ratio of NPP to GPP, which is a useful diagnostic
measure for the comparison of parameterizations for the terrestrial carbon fluxes driven by
vegetation that are implemented differently in current ESMs. The absolute magnitudes of the
production terms are the results of feedbacks between climate and vegetation. Normalized flux
terms can highlight the differences among simulations driven by parameterization differences
in terrestrial carbon fluxes. Previous studies based on in situ (De Lucia et al., 2007) and satellite
(Zhang et al., 2009) data analyses have indicated that CUE is not a constant with a value of
approximately 0.47 (Gifford, 1994; Dewar et al., 1999) but varies depending on climatic
conditions and PFTs. In this regard, the Moderate Resolution Imaging Spectroradiometer
(MODIS) satellite data provide the global coverage of GPP and NPP as a useful reference for
the model validation for CUE at the PFT level. Zhang el al. (2014) suggested observed CUE
by MODIS tends to slightly increase in the recent years.
The purpose of this study is the intercomparison of CMIP5 ESMs in terms of their
simulations of the terrestrial carbon cycle, based on a quantitative evaluation of the
performance of terrestrial carbon flux parameterizations in their land surface models (LSM).
This analysis specifically focuses on the assessment of CUE at the PFT level and makes an
effort to provide useful suggestions to the modeling community for reducing systematic biases
in the terrestrial carbon cycle in current ESMs. This study consists of following sections:
Section 2 describes the observational data and model output used in this study. Section 3
compares the model simulations in terms of their climate and terrestrial carbon cycle variables,
comparing first the multi-model ensemble (MME) average to diagnose common and systematic
biases in the current models and then identifies differences among simulations across the ESMs
in their simulated climates and carbon fluxes. The comparison of CUE at various PFT levels is



followed by more comprehensive comparisons for identifying differences among simulations
driven by model parameterizations. Finally, Section 4 provides a summary and conclusions.

**2. Data and Analysis Methods**
**2.1 Observational data**
This study used GPP and NPP as primary variables to validate the global carbon cycle as
simulated by various ESMs. Reference observational data were obtained from the NASA
MODIS MOD17 data product, which includes the first satellite-driven estimates of carbon
fluxes on vegetated surfaces on a global scale (Running and Gower, 1991; Zhao et al., 2005).
The MODIS algorithm uses a data model based on the radiation use efficiency logic of
Monteith (1972) to estimate GPP, which is basically a linear function of the amount of
Photosynthetically Active Radiation (PAR) absorbed. The fraction of PAR and the leaf area
index (LAI) are provided to the model by the MODIS MOD15 products. A conversion
efficiency parameter relating absorbed radiation to the actual productivity depends on
vegetation type and climate condition. The upper limit of conversion efficiency uses the Biome
Parameter Lookup Table (BPLUT) for different vegetation types. The vegetation types include
evergreen needleleaf forest (ENF), evergreen broadleaf forest (EBF), deciduous needleleaf
forest (DNF), deciduous broadleaf forest (DBF), mixed forests (MF), open and closed
shrublands (SHR), grasslands (GRA), and croplands (CROP), which are based on the land
cover    classification    from    the    MODIS    MCD12Q1
(https://lpdaac.usgs.gov/dataset_discovery/modis/modis_products_table/mcd12q1). Figure 1
shows the horizontal distribution of vegetation types from MODIS. The conversion efficiency
is modified by climate conditions such as incoming solar radiation, temperature, and vapor
pressure deficit, which are obtained from atmospheric reanalyses developed by NASA's Global





Modeling and Assimilation Office and the NCEP/NCAR Reanalysis II. The NPP estimation
by MODIS calculates daily leaf and fine root maintenance respiration, annual growth
respiration, and annual maintenance respiration of live cells in woody tissue, which are
subtracted from the GPP. Biome-specific physiological parameters are also specified by
BPLUT for respiration calculations.
The MOD17 dataset provides 8-day, monthly, and annual mean GPP and NPP for 2000-2012.
This study used the gridded GPP and NPP products, which have a spatial resolution of 30
arcsec (0.0083 degree), provided by the Numerical Terradynamic Simulation Group (NTSG)
of the University of Montana (NTSG MOD17 v55).
Although MODIS is affected by uncertainties in biomass types and meteorological data sets
(Zhao et al. 2005), the derived GPP and NPP values are able to capture realistic spatial and
temporal variations over different biomes and climate regimes. Zhao et al. (2005) and Heinsch
et al. (2006) demonstrated that the data are consistent with ground-based flux tower
measurements of GPP and field-observed NPP estimates with high correlation (r=0.859).
For comparison with MODIS, this study also used GPP estimates from FLUXNET-MTE
(Multi-Tree Ensemble; Jung et al., 2011), which is an upscaled data set providing global
coverage that is derived from 178 surface flux tower observations using a machine learning
technique. FLUXNET-MTE provides an explicit estimate of carbon fluxes over vegetated
surfaces. The dataset provides monthly data at a 0.5° × 0.5° (latitude × longitude) spatial
resolution and covers the period 1982 – 2007. Although this gridded global dataset is useful
for validation of ESMs, its key limitations are also discussed in the literature (Jung et al., 2011).
Wide geographical regions are not represented by measurement stations; for example, there is
a lack of samples over Siberia, Africa, South America and tropical Asia compared with North
America and Europe. Estimates of annual-mean upscaled ecosystem respiration have higher



certainty than the anomalies and show approximately 5-10 % underestimation. Additionally,
the data have limitations in accounting for disturbances due to land use changes, given that
unchanged land cover data from the International Geosphere-Biosphere Program (IGBP)
satellite are used for all periods. This may introduce spurious trends into the GPP estimates
from the FLUXNET-MTE project. The dataset does not provide estimates of Ra, but instead
provides the summation of Ra and Rh. The geographical distribution of satellite-derived GPP
from MODIS shows a high degree of consistency with that from in situ FLUXNET
observations.  Figure 2 compares the annual GPP distributions from MODIS and FLUXNET
for the same period, 2000-2005. A notable difference between the two appears in the Amazon,
where MODIS tends to underestimate the productivity significantly. In the remaining regions,
MODIS tends to produce slight underestimates in the tropics and overestimates in the high
latitudes when compared with FLUXNET. The annual GPP values from MODIS and
FLUXNET are 108.76 GtC and 107.41 GtC, respectively, for the averaging period of 2000-
2005, with a small difference that is no more than 1 % of the total value. The pattern of
differences did not change significantly even if the FLUXNET data were averaged over a
longer period (1983-2005). In fact, the interannual variation did not modify the global-mean
annual GPP value significantly when the reference period was extended to 1983-2005, which
yielded a small reduction to 106.55 GtC using the FLUXNET data.
This study also used the observed surface air temperature and precipitation data from the
Institute for Climate Impact Research based on the CRU (Climate Research Unit)
meteorological dataset (Harris et al., 2014). In this data product, temperature and precipitation
at stations worldwide were interpolated to a horizontal resolution of 0.5° × 0.5° (latitude
× longitude) covering the global land surface.




### 2.2 Model Data

Historical simulations performed using 10 ESMs were used in this study. Brief descriptions
of these models is provided in Table 1. The historical simulations (that is, experiment 5.2 or
the ESM historical 1850–2005 simulation; Taylor et al., 2012) were forced by gridded $CO_2$
emissions data for fossil fuel consumption from Andres et al. (2011). While conventional $CO_2$
concentration-driven runs have no vegetation feedback on atmospheric $CO_2$, these emissions-
driven runs enables climate-carbon cycle feedbacks via changes in vegetation. Note that three
models – GFDL-ESM2M, GFDL-ESM2G, and MPI-ESM LR – of them enabled the dynamic
vegetation model in their historical simulations for 1850 – 2005, which model was able to
consider dynamic change of PFT boundaries by climate conditions (Table 1). Atmospheric $CO_2$
concentrations are simulated prognostically from the net budget of natural and anthropogenic
carbon fluxes to and from the atmosphere. The simulation of GPP is directly controlled by the
formulae representing photosynthesis in the models. As shown in Table 1, the parameterization
of photosynthesis by vegetation is formulated similarly in the 10 ESMs. This parameterization
is mostly based on Farquhar et al. (1980) for C3 plants in cold climates, with revisions for C4
plants in warm climates by Collatz et al. (1992). Leaf photosynthesis in CLM4 is proportional
to the concentration of carbon dioxide in the atmosphere, as well as the temperature and
moisture surrounding leaves. It adjusted the minimum rate among the light-use, water-use and
carbon assimilation approaches in CLM4.
NPP is diagnosed in ESMs by subtracting Ra from GPP. Parameterizations for Ra are more
diverse in formulation across the models compared to that of photosynthesis. Note that
CESM1-BGC and NorESM-ME1 incorporate identical land surface models, in which the
nitrogen cycle is allowed to limit plant assimilation for the parameterization of carbon fluxes



by terrestrial vegetation, so called the interactive carbon-nitrogen (CN) cycle. Respiration is
proportional to temperature and nitrogen concentration. The models without interactive
nitrogen cycles diagnose nitrogen concentrations from the carbon concentration in each carbon
pool, whereas the models with interactive nitrogen cycles predict the nitrogen concentrations.
The only exception is MRI-ESM, which uses an empirical formula for estimating NPP based
on Obata (2007). In the model, the monthly NPP is empirically derived from physical variables
such as temperature and precipitation from the Miami model (Lieth, 1975; Friedlingstin et al.,

1995).

The model data were obtained from the Earth System Grid Federation (ESGF), an

international network of distributed climate data servers (Williams et al., 2011). For the
purposes of comparison, the model outputs, as well as the MODIS data, were interpolated onto
the same 1° × 1° grid (latitude × longitude).

**2.3 Analysis Methods**

In Section 3.3, CUE is diagnosed at the ecosystem level for the MODIS observations and

the various ESM simulations. For simplicity, an identical distribution of vegetated surfaces
based on to the MODIS classification (Figure 1) was applied to both the observed and the
simulated fluxes. This is because each model has their own vegetation classifications, which
are not available from the CMIP5 data archive.

It is noted that the deficiency in the simulation of CUE by individual models is not only

caused by deficiencies in the parameterization of carbon fluxes due to vegetation but also by
differences in the classifications of PFTs, which are specified differently in each model. For
example, LM3.0 in GFDL ESM2 M and ESM2G simulate 5 PFTs (i.e., 3 types of trees and 2
types of grasses), while NCAR and NorESM's CLM4.0 specifies the PFTs in much greater



detail by including 17 different types (i.e., 8 types of trees, 3 types of shrubs, 3 types of grasses
and 3 types of crops). Although referencing PFTs from the observations instead of using own
PFTs in each model might not be a perfect comparison, it is still meaningful to identify the first
order differences driven by parameterization method and the classification difference as well
where the latter is regarded as the model bias too.


**3. Results**
**3.1. Systematic Biases in the Multi-Model Ensemble**
Systematic biases in the ESM simulations are examined first by taking multi-model
ensemble averages (MME) for simulated surface air temperature and precipitation, respectively
(Figure 3). Despite the realistic representation of annual-mean surface temperatures, MME
exhibits systematic biases with significant hemispheric differences. Warm biases are seen in
the Northern Hemisphere, particularly in northeastern Asia and North America, whereas there
exists a cold bias in most of the Southern Hemisphere. MME generally shows wet biases in
precipitation, except over South America. Wet biases seem to be consistent with cold biases in
the tropical regions, where the deep convective rainfall tends to produce deep clouds that
attenuate incoming solar radiation at the surface.
The annual GPP, NPP and Ra values from the MODIS observations and the MME are
compared in Figure 4. The observed GPP values from MODIS are generally high in areas of
EBF in tropical regions, such as Amazon, South Asia, and Central Africa, and in areas of DBF,
such as those in Indochina, China, India, Europe and the southeastern part of North America.
GPP is observed to be small in areas of SHR in Australia and in boreal regions of MF and GRA
in northern Eurasia. GPP is close to zero over dry and non-vegetated surfaces, such as the





Sahara Desert and central Australia. The MME of the ESMs tends to reproduce these
geographical differences realistically, although the estimated magnitudes are too large over
most of the globe. Although Ra tends to be overestimated as well, MME shows a net positive
bias in NPP in most terrestrial regions, suggesting that the MME should underestimate the
observed trend of atmospheric $CO_2$ increase.
The global-mean values of GPP, NPP, and Ra are compared in Figure 5. Note that spread of
the simulations is large, particularly due to the outlier value produced by MRI-ESM1. The
median value of GPP simulated by ESMs is centered slightly above the value from MODIS
and is approximately 20 % higher (+18 GtC). The median value of NPP is also overestimated
by 10.2 GtC compared with the 52.1 GtC NPP from MODIS. The median value of Ra is
underestimated.
The formulations of GPP and Ra are closely related to temperature and precipitation
(Rahman et al., 2005; Yang et al., 2006), and, the model biases in those carbon fluxes might be
driven both by systematic biases in climate conditions such as temperature and precipitation
and the uncertainty in the parameterization formulations themselves. The Taylor diagram is a
common and useful measure for simulated spatial distributions that calculates spatial
correlation coefficients between observed and simulated values and the normalized standard
deviation of simulated values from the global mean over the whole domain of comparison.
Figures 6a and 6b show Taylor diagrams for the annual mean surface air temperature and
precipitation, respectively. The MME simulation of temperature by the CMIP5 ESMs is quite
close to the CRU observations. The spatial correlations are greater than 0.95 in all models. The
normalized standard deviations are within the range of 0.8 to 1.5, which is relatively small
compared with other simulated variables. The Taylor diagram of precipitation shows less
accuracy and more model spread than that of SATs. The spatial correlation of the MME is





approximately 0.76; the MME also shows higher normalized standard deviations compared
with temperature, suggesting that current ESMs exhibit relatively larger discrepancies in
precipitation and the terrestrial water cycle. Spatial patterns of GPP simulated by the ESMs
(Figure 6c) show even larger systematic biases with lower spatial correlations and larger spatial
changes (i.e., higher normalized standard deviations) than the observed values. Model spread
becomes much larger than that of temperature and precipitation. The simulated pattern
correlations from the ESMs are lowest for NPP (Figure 6d). The correlation for the MME is
slightly higher than 0.5. The models also exhibit much higher spatial variation than the
observed values for both GPP and NPP.
The Taylor diagram analysis suggests that the systematic biases in the ESMs may be
successively amplified by deficiencies in the simulation of climate and the terrestrial carbon
cycle. Regarding the climate conditions that affect the terrestrial carbon cycle, particularly the
distribution of precipitation and the water cycle seem to contribute more to the bias than does
temperature. In addition, the much larger spread in GPP and NPP simulated by the ESMs
compared to that in temperature and precipitation suggests that there should be much larger
uncertainty in the parameterization of terrestrial carbon cycle in the current ESMs. Biases and
model spread are even larger in NPP compared with GPP, implying that the simulation
uncertainty is much larger when the photosynthesis and the respiration are combined. The
performance of the MME in terms of GPP and NPP is not necessarily higher than that of the
individual models in this case, due to the presence of persistent and large deficiencies in the
individual models.

**3.2. Model Dependences**
The simulation of annual GPP values shows significant model dependence as shown in



Figure 5. MRI-ESM1 shows the largest value among the models. The three models, ESM2G,
ESM2 M, and MPI-ESM-LR, simulate relatively larger values of GPP than the rest of the
models. As the simulation of Ra shows relatively small model dependence, models that
simulate larger GPP values tend to produce larger NPP in general. MRI-ESM1 is an exception,
and the simulated GPP of this model is significantly reduced by its large Ra, leading to an NPP
value close to the median value. The two models, CESM1-BGC and NorESM1-ME, that share
the same land surface model simulate the smallest NPP values, which is a significant
underestimation relative to the MODIS estimate.
To examine further what causes the global bias in carbon fluxes, the spatial distribution of
the GPP bias pattern in carbon fluxes simulated using each model is compared in Figure 7.
Each model exhibits its own systematic biases. MRI-ESM1 shows a significant positive bias
in most vegetated regions, which is particularly pronounced in tropical rainforests. The group
of models with higher global-mean GPP values in Figure 5 (i.e., MPI-ESM1-LR, ESM2 M,
and ESM2G) shows GPP bias patterns that are remarkably similar to each other. GPP is
overestimated in most regions in these models except for the upper inland region of the Amazon.
The rest of the models show mixed spatial patterns of positive and negative biases. The large
negative GPP bias in part of the Amazon is primarily responsible for the lowest global-mean
GPP values, which are simulated by CanESM2 and BCC_CSM1 M. The negative bias is clear
in the boreal high-latitude regions above 40 N in the CESM1-BGC and NorESM1-ME models.
The systematic biases in the models reflect the uncertainties in the parameterized carbon cycles,
as well as in the simulated climates. Most models simulate larger production in the tropics, due
to abundant rainfall and high temperatures, and smaller production in high latitudes due to less
precipitation and low temperatures. As GPP is much larger in magnitude than Ra, the NPP bias
pattern in each model is mostly dominated by that of GPP rather than Ra, leading to consistent



patterns (cf. Figure 7 and Figure 8). The two GFDL models implemented with the same LM3
land surface model (i.e., ESM2M and ESM2G) and the other two models that use CLM4
(CESM1-BGC and NorESM1-ME) show NPP biases with opposite signs in the boreal regions
above 40 N, highlighting significant model differences in parameterizations of carbon fluxes
due to vegetation.

**3.3. Carbon Use Efficiency**
The bias patterns of GPP and NPP simulated by the various ESMs presented in Figure 7 and
8 are the result of complicated feedbacks between the carbon cycle (mostly by terrestrial
vegetation) and climate. As the magnitude of the bias is also a function of biomass, this study
further compared carbon use efficiency by dividing NPP by GPP. This normalized carbon flux
ratio can highlight the difference among simulations driven by parameterization differences in
terrestrial carbon fluxes by vegetation. The spatial pattern of CUE obtained by MODIS shows
significant variations (Figure 9). In MODIS, most tropical areas with high GPP values
generally show low CUE values below 0.4, particularly over the Amazon, central Africa and
Southeast Asia. In contrast, CUE is in general greater than 0.5 over wide areas in high latitudes
and a few low-latitude, high-elevation regions. The spatial distribution of CUE apparently
depends on climate conditions such as precipitation and temperature in that regions with large
amounts precipitation and warm climates show low CUE values, while regions experiencing
dry and cold climates show high CUE values. Overall, the MME of 10 ESMs tends to reproduce
the observed distribution from MODIS reasonably well. However, the MME values are lower
than the observed values in most regions, which can largely be attributed to the underestimation
of CUE values by MRI-ESM1. The bias pattern of CUE differs strongly among the models.
Note that the bias pattern of CUE tends to characterize the parameterization differences in the





terrestrial carbon fluxes used in the ESMs. The bias patterns of CUE are almost identical to
each other for models that share the same land surface model, such as BCC_CSM1 and
BCC_CMS1 M, and ESM2 M and ESM2G, and CESM1-BGC and NorESM1-ME,
respectively. The two BCC models tend to overestimate CUE in Eurasia, North America, and
Africa, while they produce underestimates in Australia and South America. CanESM2 shows
a similar pattern as the two BCC models. MPI-ESM1-LR shows a similar bias structure except
in that it produces overestimates in South America. CESM1-BGC, NorESM1-ME, and MRI-
ESM1 exhibit an underestimation of CUE over most terrestrial regions.

The model dependence is depicted better by the zonal mean CUE distribution (Figure 10).

The observed CUE values show a clear latitudinal dependence and generally increases with
latitude. The zonal mean of CUE from MODIS ranges from 0.3 to 0.7, with a global average
of 0.49. It indicates that the biomass in high latitudes tends to take up atmospheric carbon more
efficiently compared with that in tropics. Even though the model spread is larger, the zonal
mean MME is able to reproduce the observed relationship between CUE and latitude. Some
models, such as CESM1-BGC, NorESM1-ME and MRI-ESM1, are notably different from the
other models, as well as from MODIS, and simulate low values, particularly at middle to high
latitudes. These results are consistent with those in Shao et al. (2013). They suggested that
respiration decreases more rapidly than production in response to latitudinal decreases in mean
temperature in all models expect NorESM1-ME and CESM1-BGC. The reason for the
underestimation of CUE in the two models are caused by their low estimates of NPP. Using the
same data from MODIS, Zhang et al. (2009) suggested that there exists a clear relationship
between CUE and climate conditions, such as surface air temperature and precipitation, that
are critical for biomass growth.

Figure 11 compares the relationship from MODIS with the model simulations. The observed



CUE from MODIS is more influenced by temperature than precipitation, as is particularly clear
in dry regions with precipitation below 50 mm yr$^{-1}$. In general, the observed CUE decreases
with increasing temperature. Moreover, observed CUE values show the sensitivity of CUE to
precipitation in the tropics, where plant growth is more sensitive to precipitation compared
with high latitudes. The MME basically follows this temperature sensitivity, although it tends
to underestimate CUE. It is caused by the overestimation of Ra in most models compared with
the MODIS estimates (Figure S3). Individual models show their own deficiencies. For example,
the GFDL models (ESM2 M and ESM2G) tend to overestimate the sensitivity of CUE to
precipitation in tropical regions compared with MODIS. It indicates that the gradients in CUE
with temperature in the GFDL models are weaker than those in MODIS. In contrast, the models
based on CLM4.0, such as CESM1-BGC, NorESM1-ME and MRI-ESM1, show a weaker
sensitivity of CUE to both temperature and precipitation than the other models. This result
might be caused by other limiting and trigger processes, such as nitrogen limitation, which are
larger than the sensitivity to temperature and precipitation. This large divergence in the model
sensitivity of CUE to temperature and precipitation induces differences in the atmospheric CO2
concentrations in the future among the full coupled ESMs.
Figure 12 compares the observed values and differences among simulations in terms of CUE
depending on the dominant PFTs according to the classification in Figure 1. In the MODIS
observations, the CUE values over broadleaf forests (DBF and EBF) are generally lower than
over needleleaf forests (DNF and ENF), implying that dense forests tend to not only take up
large amounts of atmospheric carbon for photosynthesis but also release large amounts of
carbon to the atmosphere though respiration. In this regard, the efficiency of carbon uptake by
the broadleaf forests is smaller than that of needleleaf forests.
The observed variations in CUE depending on the PFTs are reproduced realistically by the



MME. The differences between MODIS and the MME is large in areas of DNF and DBF, but
those vegetation types occupy relatively small fractions of the vegetated surface. The model
spread is large, regardless of plant function types. This is primarily due to the low CUE values
produced by three of the models, CESM1-BGC, MRI-ESM1 and NorESM1-ME, for all of the
plant function types. These three ESMs have their own unique formulations in parameterizing
terrestrial carbon fluxes. In the case of MRI-ESM1, it determines the monthly Ra empirically
based on a function of the surface air temperature and precipitation (Obata, 2007). The
simulated NPP in MRI-ESM1 is the residual term between GPP and Ra that is evidently
different from that of the other ESMs. The two CLM 4.0-based models, CESM1-BGC and
NorESM1-ME, include coupled carbon and nitrogen (CN) cycles, which seems to lead to
dramatic differences in CUE compared with the other models that do not represent interactions
between the carbon and nitrogen cycles. Inclusion of the nitrogen cycle in the models tends to
constrain the amount of carbon uptake in vegetated land surface (Zaehle et al., 2010;
Friedlingstein et al., 2014) and produces higher simulated growth respiration than in other
models (Shao et al., 2013).
To examine the impact of the CN cycle in the model further, this study conducted two
additional sensitivity experiments using CESM1-BGC, one with interactive carbon-nitrogen
cycle (CN) and the other with no nitrogen cycle (Only C). Figure 13 shows that CN tends to
decrease GPP in most of areas compared with Only C, whish suggests that the implementation
of nitrogen cycle in this model reduces the amount of carbon uptake by vegetation drastically
as a limiting factor. Accordingly NPP also tends to decrease in most of the regions at the
decrease of GPP. It is interesting to see that CUE decrease is particularly significant in mid- to
high-latitudes rather than in the tropics. This result is quite consistent with the simulation
difference between the CN models (CESM1-BGC and NorESM1-ME) and the rest of ESMs





(e.g., the zonal mean CUE shown in Figure 10).

This study further compares the observed and the MME-simulated CUE sensitivity to the

surface temperature for each plant function type (Figure 14). The MODIS observations show
more scatter in CUE values for a given temperature, suggesting that the natural carbon cycle is
not simply determined by temperature, but is also controlled by other factors. In most PFTs,
the observed CUE is maintained close to or even higher than 0.6, particularly in low canopy
plants such as SHR, CROP and GRA, for surface temperatures lower than 10 ℃. CUE tends
to decrease significantly at temperatures higher than 10 ℃. This observed feature may be
interpreted based on the ecological significance of the resistance to low temperatures by plants
(Allen et al., 2010). Low temperatures tend to reduce biosynthetic production by plants and
can even disturb vital functions to cause permanent injuries and death. The survival capacity
of plants tries to make its metabolic processes continue to function under low temperature
stresses and using its cold resistance (Larcher, 1968). It suggests that the CUE values of
vegetation may be lowered in favorable environmental conditions, such as warm temperatures
and abundant precipitation, as there is plenty of production and plant growth. Vegetation
experiencing cold temperatures and insufficient precipitation adapts to survive by increasing
CUE.

In contrast, even though the multi-model ensemble average is taken for the various ESMs,

the simulated CUE variation shows a clearer change with temperature, suggesting that the
parameterization of the terrestrial carbon cycle in current ESMs depends too much on
temperature conditions. A decreasing trend is clear in the MME regardless of PFTs in response
to an increase in temperature. From the MME simulation results, CUE values in all PFTs shows
a clear linear change in response to temperature variation. This implies that the current models




do not adequately consider the observed ecological resistance to temperature, and the balance
between respiration and production in the models is more simplified than the observations. In
fact, the parameterizations of most land surface models are based on conceptual leaf-level
formulations, such as those used in the calculation of biochemical photosynthesis processes
and the dependence of $CO_2$ exchange on stomatal conductance, which use temperature and soil
moisture explicitly in their formulations. The comparison results in this study suggest that the
models might need to consider ecosystem-level parameterizations which simulate carbon and
nitrogen fluxes and vegetation and soil pools and are estimated at a long (e.g., monthly) time
step based on spatially explicit information on climate, ecosystem type, soil type, and elevation
(Zhu and Zhuang, 2015) to reflect the nonlinear relationship for the interaction between climate
condition and vegetation.

**4. Summary and Concluding Remarks**
The simulations of climate and the terrestrial carbon cycle have been examined by comparing
surface temperatures and precipitation, as well as GPP, Ra, and NPP values, simulated by 10
different CMIP5 ESMs with the CRU surface observational data for climate-related variables
and the MODIS satellite estimates for the carbon cycle over 6 years (2000-2005).
Despite the systematic biases with significant hemispheric differences, the spatial
distributions of temperature and precipitation, which are closely related to biogeochemical
variables (Rahman et al., 2005; Yang et al., 2006), are relatively similar when compared with
observations. More model discrepancies appeared in the simulation of the carbon cycle, which
reflects overestimation of GPP over most of the globe. The terrestrial carbon fluxes simulated
by the ESMs are diverse, and the models exhibit large spread, even though the multi-model
ensemble mean (MME) shows strong resemblance in terms of its spatial distribution to the





observed pattern by cancelling out the systematic biases in each model. The results show that
the biases of terrestrial carbon fluxes are due less to the bias in the spatial distribution of climate
conditions but more to the larger uncertainty in their parameterizations.
We also analyzed carbon use efficiency (CUE) by dividing NPP by GPP, which is a
physiological parameter defined as the proportion of carbon acquisition (e.g., GPP) to
vegetation growth (NPP). Analyzing CUE help us to understand the carbon storage in
simulated terrestrial ecosystem in ESMs. At first, the spatial distribution of observed CUE from
space (e.g., MODIS) depends on climate condition such as precipitation and temperature. For
example, the regions of large precipitation and warm climate show low CUE, while the regions
of dry and cold climate show high CUE. It indicates that CUE at the regions with warm
temperature and abundant precipitation could be lowered as there is a plenty of production and
plant growth. The vegetation in cold temperature and insufficient precipitation adapts to the
environmental condition for survival by increasing CUE.
In different with MODIS, we found clear difference of CUE between ESMs. The bias pattern
of two ESMs from BCC showed the hemispheric contrast to positive in NH and negative in
SH. The strong negative bias of CUE over southern hemisphere is shown in GFDL's models.
The CUE in ESMs based on CLM4 (e.g., CESM-BGC and NorESM-ME) are significantly
underestimated globally. This large uncertainty of CUE in individual models is influenced by
biogeochemical parameterization of land surface model. In the MME, the spatial distribution
of CUE is reasonably simulated. However, Strong negative bias is found over Amazon. It is
caused that unbalanced ratio of GPP and Ra in the terrestrial carbon fluxes over tropical forest
such as evergreen broadleaf forest the most models. The inverse relationship between
temperature and CUE is reasonably simulated in the MME over dry regions. Generally, Ra is
more sensitive to temperature than GPP in the real world over a certain range of temperatures





(Woodwell et al., 1990; Ryan, 1991; Piao et al., 2010). It means that the sensitivity of
temperature to photosynthesis is weaker than that of respiration (Arnone and Korner, 1997;
Enquist et al., 2007).
The CUE variation depending to the PFTs, MME is realistically reproduced in every PFTs.
The model spread is large. It indicates a wide spread due to the different PFTs in each land
models and systematic bias such as failure of PFT description in land models. The observed
CUE values show a reasonable degree of non-linearity in terms of its response to temperature.
In contrast, the stronger sensitivity of CUE to temperature increases in the MME is reflected
by the systematic biases of simulated biogeochemical processes which depends on temperature
conditions strongly in every PFTs.
However, most of the advanced ESMs have adopted leaf-scale biogeochemistry which
involves parameterizations of photosynthesis and respiration based on small spatio-temporal
scales that depend on laboratory experiments and limited in situ studies. It makes up one of the
major uncertainties of carbon cycle processes in future climate change simulations from recent
advanced ESMs. Atkin et al. (2008) suggested that most biogeochemical models are adjusted
and incomplete parameterizations of biogeochemical processes. Due to the lack of
observational data, many biogeochemical studies have focused on the total amount of primary
production and respiration. Therefore, understanding and evaluating the global-scale
ecosystem is challenging, based on the leaf scale biogeochemical parameterization used in the
models. This leaf-level parameterization for biogeochemical processes is insufficient for long-
term simulations (Zaehle et al., 2014). For realistic long-term simulations, such as climate
change experiments including the carbon cycle and feedback processes, parameterizations
representing idealized and generalized ecosystem-level processes are needed, rather than site-
specific and leaf-level processes.





**Acknowledgement**

This study is supported Basic Science Research Program through the National Research Foundation of Korea (NRF), funded by the Ministry of Education, Science and Technology (2012M1A2A2671851) and the Supercomputing Center/Korea Institute of Science and Technology Information with supercomputing resources including technical support (KSC-2015-C3-035).



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



**Table 1. List of ESMs used in this study and their features**

| Number | Models | Modeling center | Horizontal resolution | ESM Reference | Land model | Photosynthesis | Autotropic Respiration | Nitrogen Cycle | Dynamic Vegetation |
|---|---|---|---|---|---|---|---|---|---|
| 1 | BCC-CSM1 | Beijing Climate Center, China | 2.812° × 2.812° | Wu et al. (2013) | BCC-AVIM1 | Farquhar et al., (1980) | Foley et al. (1996) | No | No |
| 2 | BCC-CSM 1M | Beijing Climate Center, China | 1.125° × 1.125° | Wu et al. (2013) | BCC-AVIM1 | Farquhar et al., (1980) Collatz et al. (1992) | Foley et al. (1996) | No | No |
| 3 | CanESM2 | Canadian Centre for Climate Modeling and Analysis, Canada | 2.812° × 2.812° | Arora et al. (2011) | CTEM | Farquhar et al., (1980) Collatz et al. (1992) | Ryan (1991) | No | No |
| 4 | CESM1-BGC | Community Earth System Model Contributors, NSF-DOE-NCAR, USA | 1.25° ×0.9° | Long et al. (2013) | CLM4 | Farquhar et al., (1980) Collatz et al. (1992) | Foley et al. (1996) | Yes | No |
| 5 | GFDL-ESM2M | NOAA Geophysical Fluid Dynamics Laboratory, USA | 2.5° ×2° | Dunne et al. (2013) | LM3 | Farquhar et al., (1980) Collatz et al. (1992) | Foley et al. (1996) | No | Yes |
| 6 | GFDL-ESM2G | NOAA Geophysical Fluid Dynamics Laboratory, USA | 2.5° ×2° | Dunne et al. (2013) | LM3 | Farquhar et al., (1980) Collatz et al (1992) | Ryan (1991) | No | Yes |




| 7 | MIROC-ESM | Japan Agency for Marine-Earth Science and Technology, Atmosphere and Ocean Research Institute, and National Institute for Environmental Studies, Japan | 2.812° × 2.812° | Watanabe et al. (2011) | MATSIRO+ SEIB-DGVM | Farquhar et al., (1980) | Ryan (1991) | No | No |
|---|---|---|---|---|---|---|---|---|---|
| 8 | MPI-ESM LR | Max Planck Institute for Meteorology, Germany | 2.812° × 2.812° | Ilyina et al. (2013) | JSBACH | Farquhar et al., (1980) | Obata (2007) | No | Yes |
| 9 | MRI-ESM1 | Meteorological Research Institute, Japan | 1.125° ×1.125° | Yukimoto et al. (2011) | HAL | Farquhar et al., (1980) Collatz et al. (1992) | Ryan (1997) | No | No |
| 10 | NorESM1-ME | Norwegian Climate Centre, Norway | 2.5° ×1.875° | Tjiputra et al. (2013) | CLM4 | Farquhar et al., (1980) Collatz et al. (1992) | Foley et al. (1996) | Yes | No |










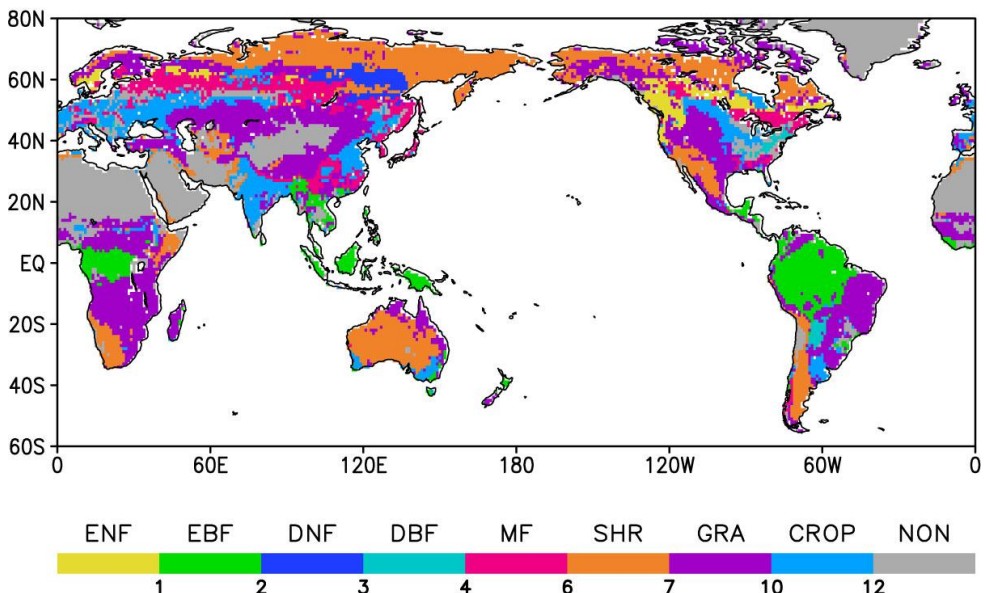


**Figure 1**. Horizontal distribution of dominant plant function types (PFTs) using the MODIS
land cover data that include evergreen needleleaf forest (ENF), evergreen broadleaf forest
(EBF), deciduous needleleaf forest (DNF), deciduous broadleaf (DBF), mixed forest (MF),
shrub land (SHR), grass (GRA), cropland (CROP) and non-vegetated area (NON).
















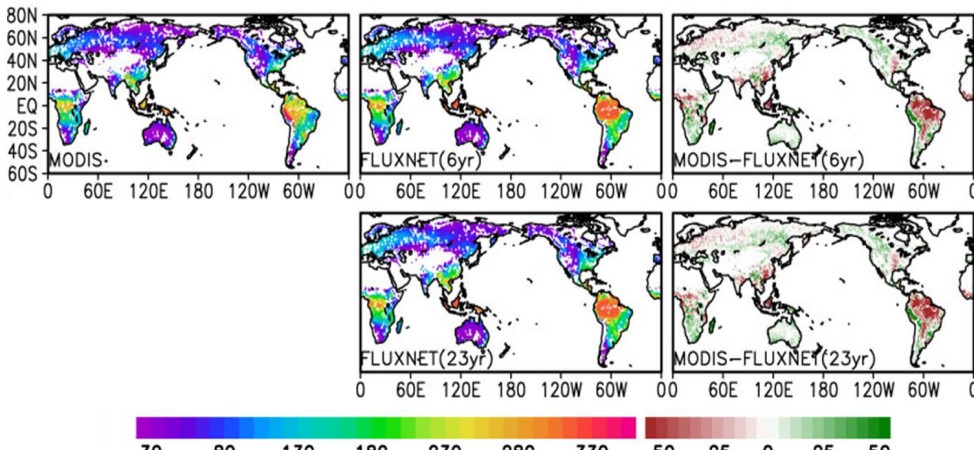


**Figure 2.** Spatial distributions of annual-mean GPP from MODIS (upper left), FLUXNET
(upper middle), and MODIS minus FLUXNET (upper right) averaged for 6 years (2000-2005).
Bottom panels show the GPP from FLUXNET averaged for 23 years (1983-2005, bottom left),
and its difference from MODIS averaged for 6 years (bottom right). The unit is gC m$^2$ mon$^{-1}$.













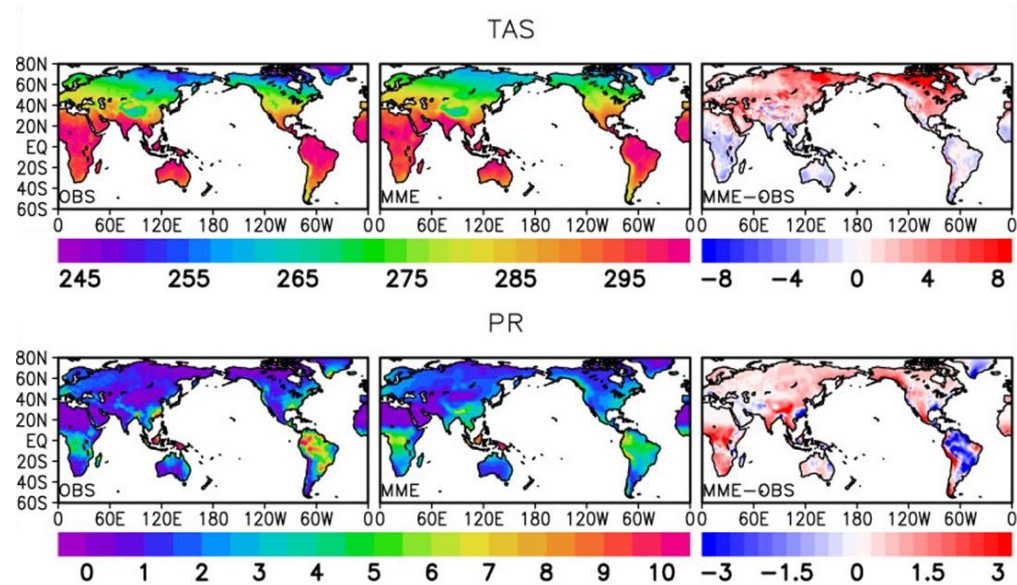


**Figure 3**. Annual-mean surface air temperature (top panels, unit: K) and precipitation (bottom

panels, mm d$^{-1}$) averaged for 2000-2005 from the CRU observations (left), and the multi-model

ensemble (MME) mean (middle), and the model biases (MME minus CRU, right).





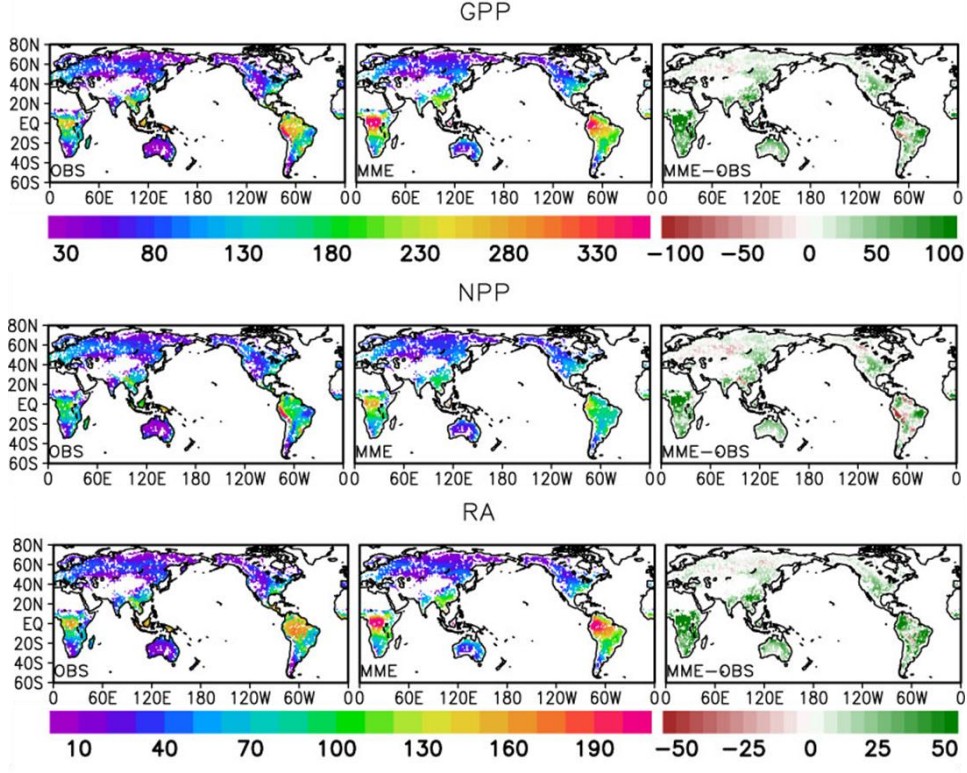


**Figure 4**. Same as in Figure. 3 except GPP (top), NPP (middle), and Ra (bottom) from the

MODIS observations and MME. The unit is gC m$^2$ mon$^{-1}$.
















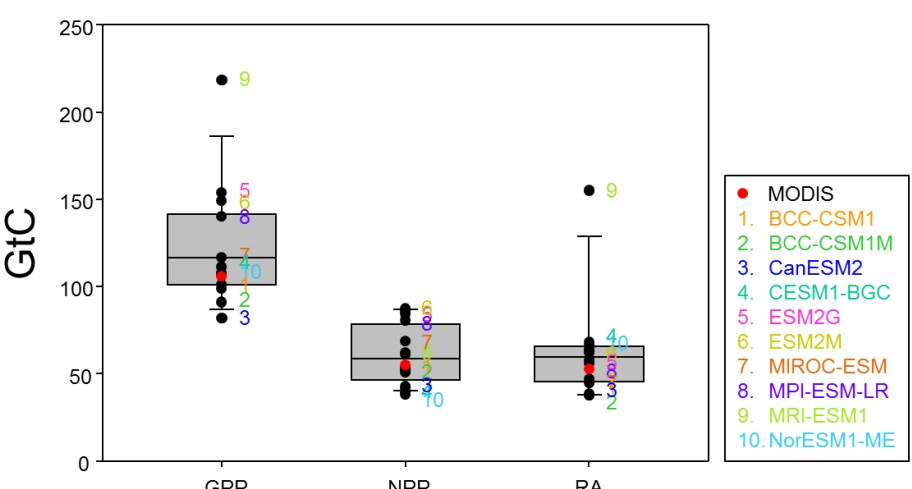


**Figure 5**. Global-mean values of GPP, NPP and Ra from MODIS and CMIP5 ESMs. The

values are the average over the land grids only with latitude weighting for the period of 2000

− 2005.



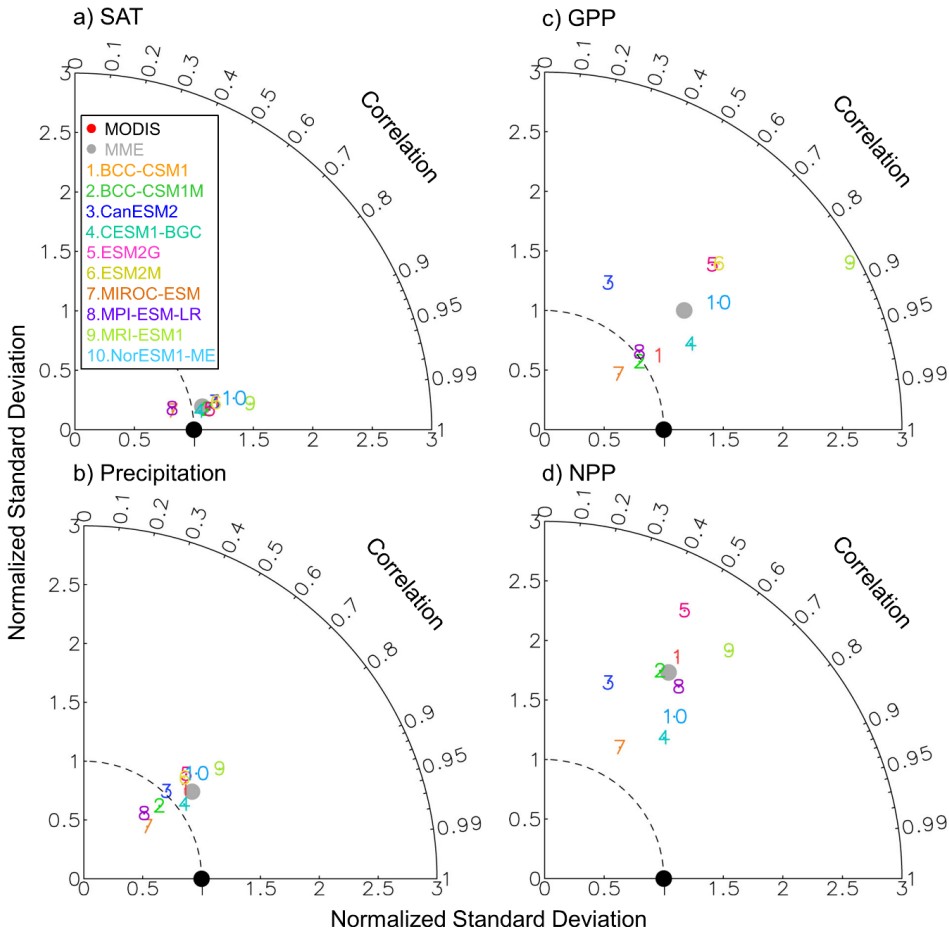


**Figure 6**. Taylor diagram of CMIP5 ESMs for annual-mean distribution of (a) surface air

temperature, (b) precipitation, (c) gross primary production (GPP) and (d) net primary

production (NPP) with respect to the corresponding observations for 6 years (2000-2005). Only

the vegetated grid points were included. The observed values are from CRU for temperature

and precipitation are MODIS for GPP and NPP.









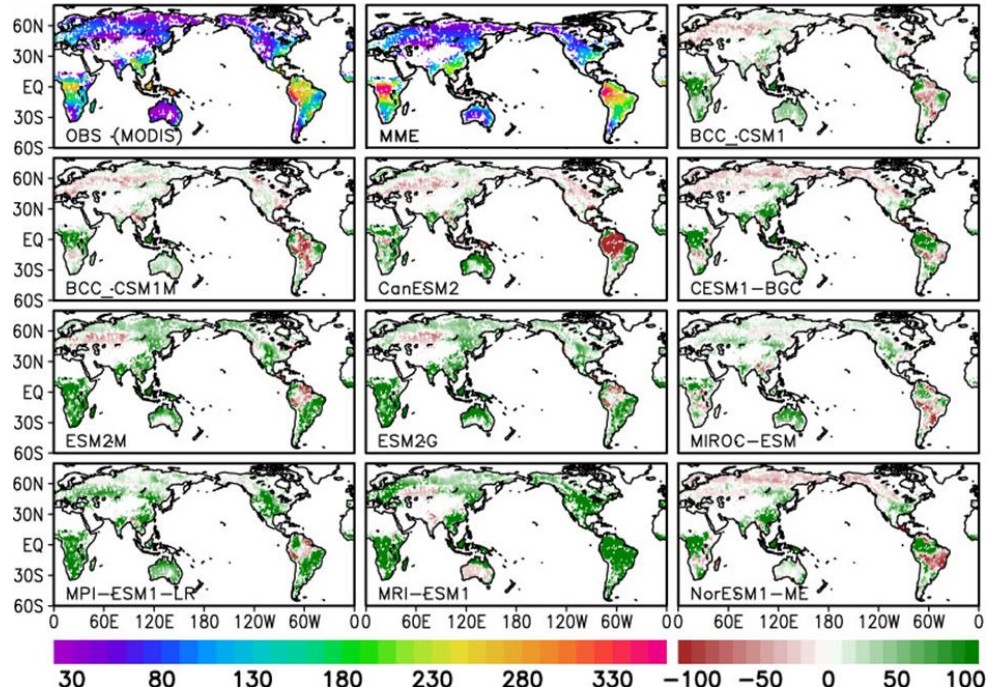


**Figure 7**. Spatial distribution of annual GPP from the MODIS observation (top left), MME

(top middle) and the simulation bias in each model (model minus MODIS). The unit is gC m$^2$

mon$^{-1}$.






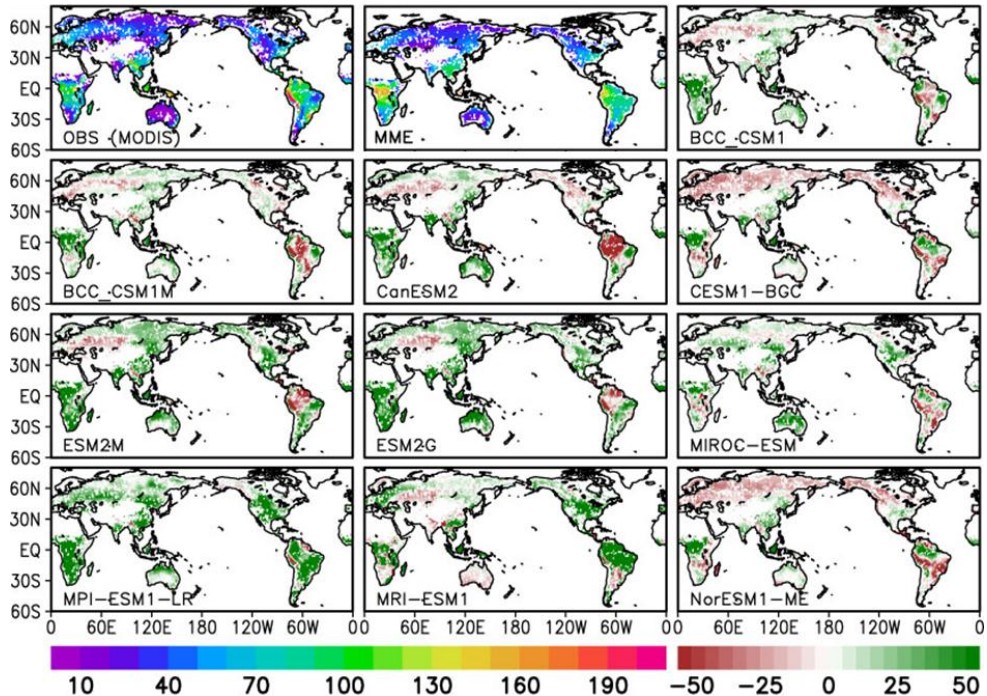


**Figure 8**. Spatial distribution of annual NPP from the MODIS observation (top left), MME (top middle) and the simulation bias in each model (model minus MODIS). The unit is gC m$^2$ mon$^{-1}$.














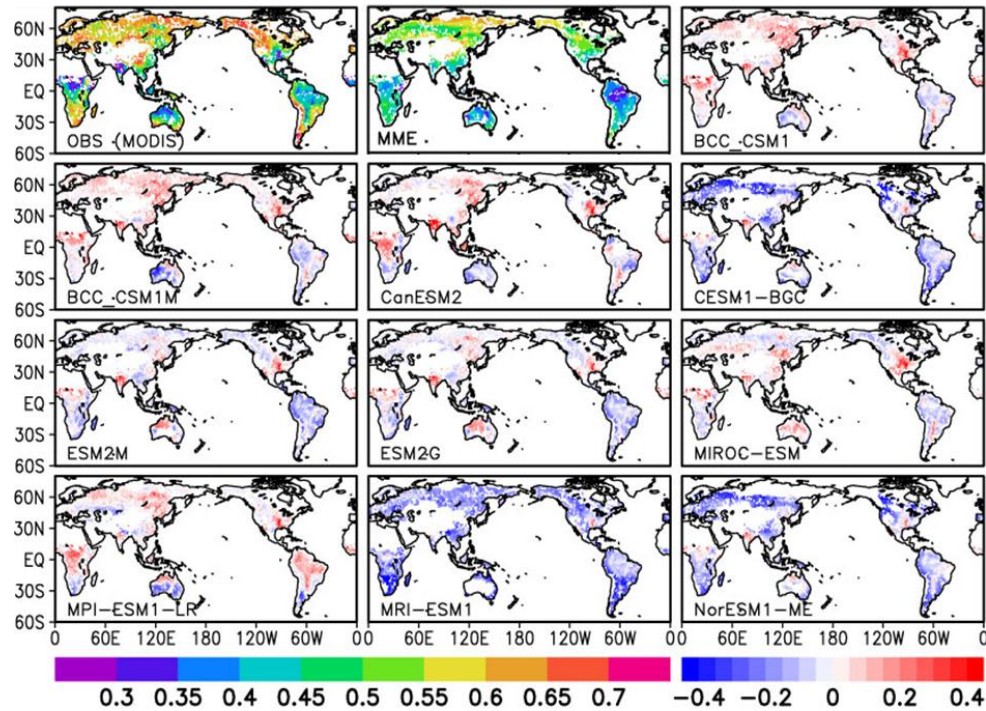


**Figure 9**. Spatial distribution of annual CUE from the MODIS observation (top left), MME

(top middle) and the simulation bias in each model (model minus MODIS). CUE is a

positively-defined ratio as NPP divided by GPP and less than or equal to 1.
















**Figure 10**. The zonal mean CUE from MODIS (black), MME (grey), and 10 ESMs (grey

circles with number).
















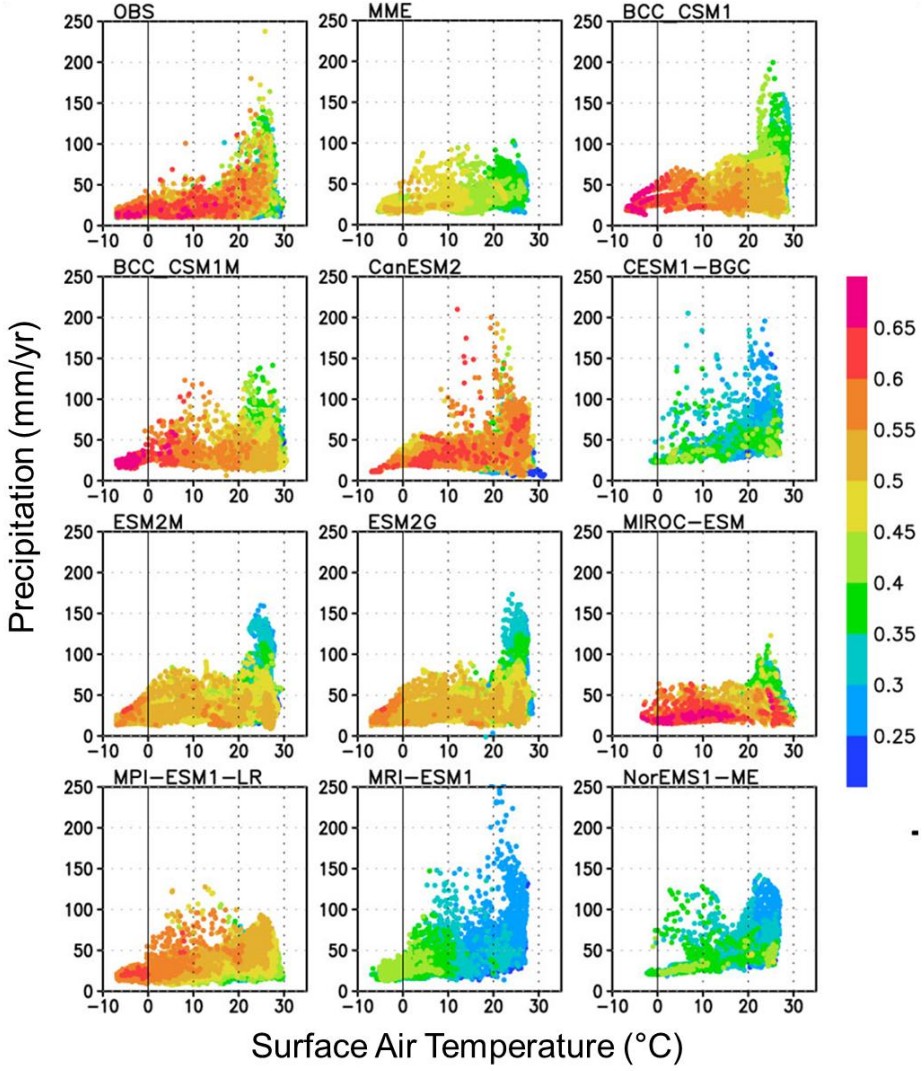


**Figure 11**. Scatter plot of CUE with the variation of surface air temperature (x-axis) and

precipitation (y-axis). Color indicates CUE.













**Figure 12**. CUE averaged for each PFT. The box widths are proportional to the root mean square of number of grids. The coefficients of proportionality box widths in each PFTs are: ENF (0.80), EBF (0.48), DNF (0.12), DBF (0.11), MF (1.25), SHR1 (0.91), SHR2 (1.78), GRA (0.70) and CROP (0.73).













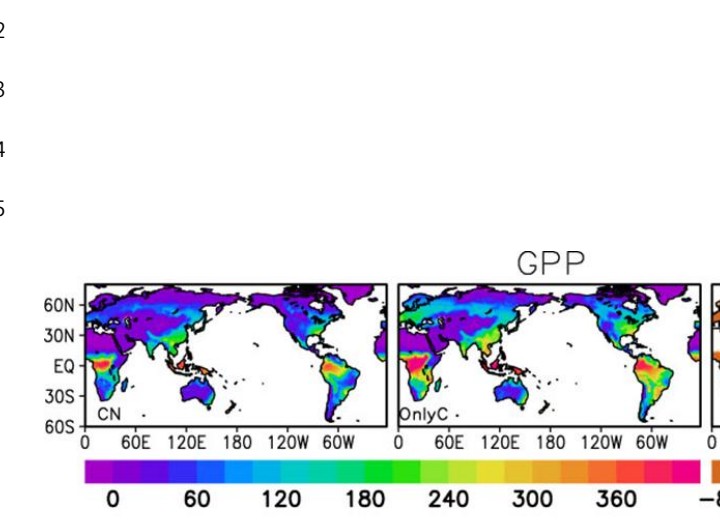

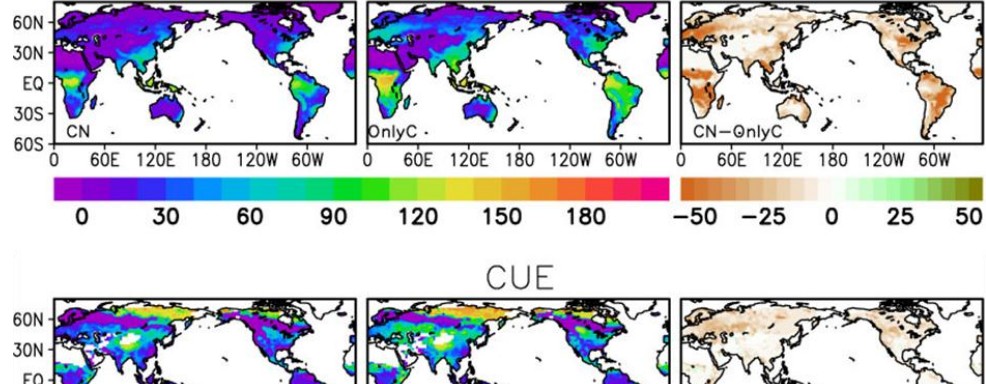

**Figure 13**. Spatial distributions of annual GPP, NPP and CUE and their differences from the

interactive carbon-nitrogen cycle simulation (CN) and the run with no nitrogen cycle (Only C)

by CESM-BGC. The units of GPP and NPP are gC m$^2$ mon$^{-1}$. CUE is a positively-defined ratio

as NPP divided by GPP and less than or equal to 1.






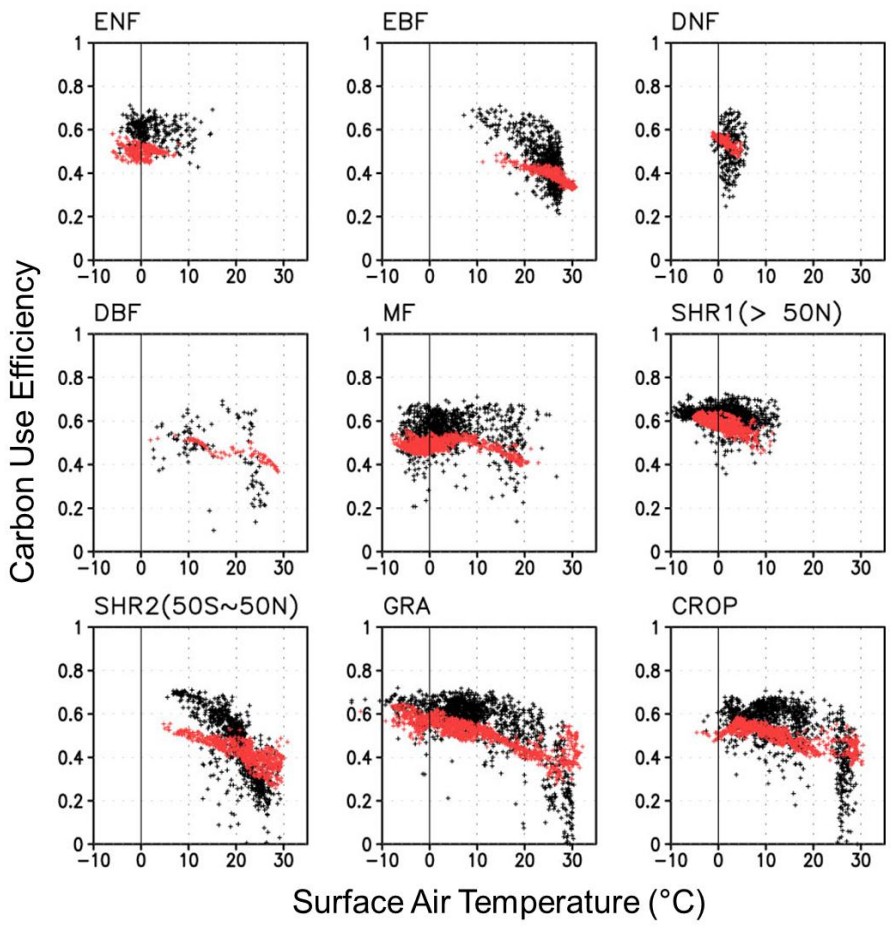


**Figure 14**. Scatter plots of CUE (y-axis) as a function of temperature (x-axis). Each panel

shows the plot for different PFT. Satellite-derived values from MODIS are presented with black

dots and the multi-model ensemble (MME) means by 10 ESMs are with red dots.









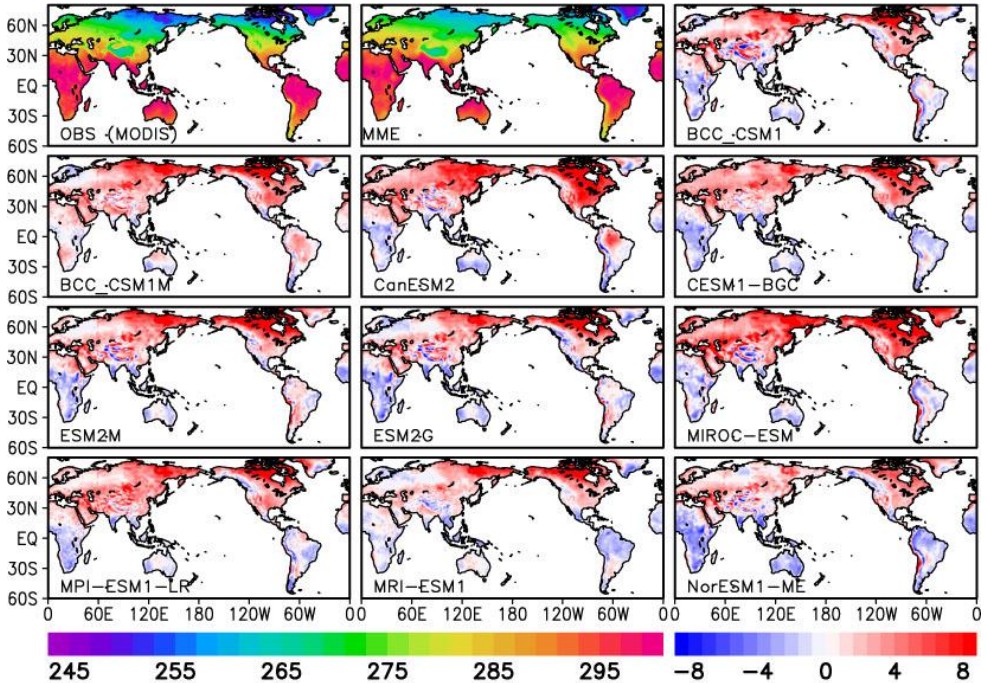


**Figure S1.** Spatial distribution of annual-mean surface air temperature from the CRU observation (top left), MME (top middle) and the simulation bias in each model (model minus CRU). The unit is K.


















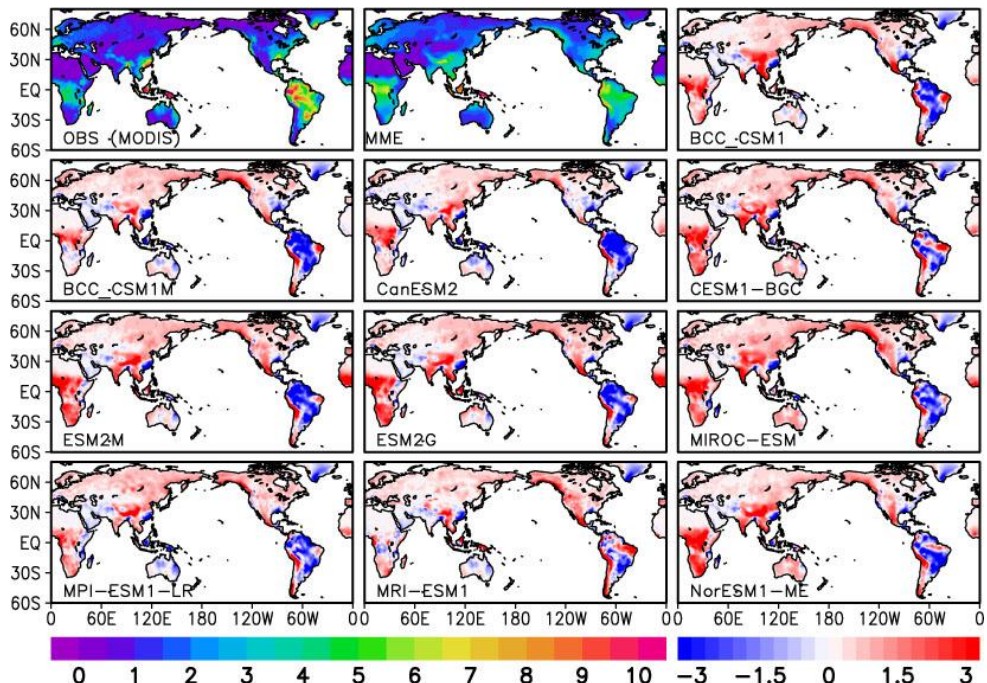


**Figure S2.** Spatial distribution of annual-mean precipitation from the CRU observation (top

left), MME (top middle) and the simulation bias in each model (model minus CRU). The unit

is mm d⁻¹.




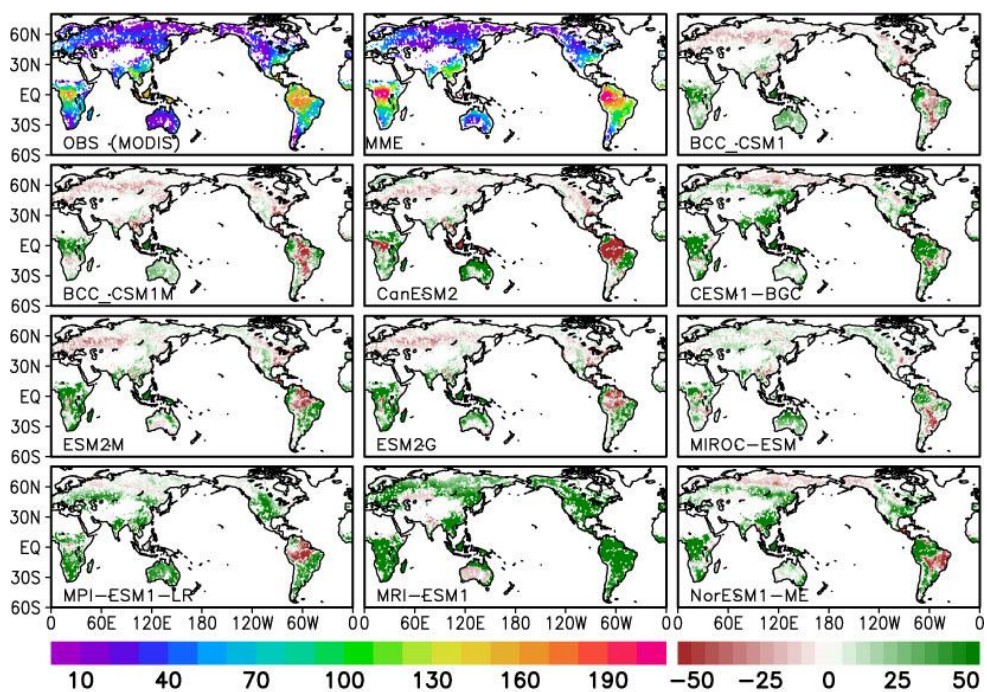

**Figure S3**. Spatial distribution of annual Ra from the MODIS observation (top left), MME (top middle) and the simulation bias in each model (model minus MODIS). The unit is gC m$^2$ mon$^{-1}$.





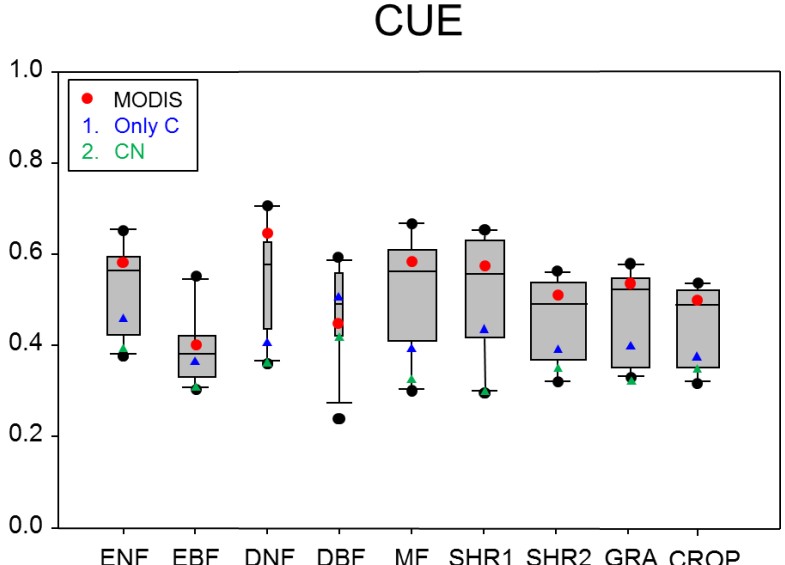

**Figure S4.** Simulated CUE averaged for each PFT in the two model sensitivity experiments
using NCAR CEMS-BGC. CN (green triangles) indicates the run with interactive CN cycle
and Only C (blue triangles) indicates the run that the nitrogen limitation effect is disabled.
MODIS is also shown in red dots. The box widths are proportional to the root mean square of
number of grids. The coefficients of proportionality box widths in each PFTs are: ENF (0.80),
EBF (0.48), DNF (0.12), DBF (0.11), MF (1.25), SHR1 (0.91), SHR2 (1.78), GRA (0.70) and
CROP (0.73).