# Peer review of "Published: 16 December 2016"

_Biogeosciences, 2016_

## Referee Comment (RC1) · Anonymous Referee #1 · 25 Jan 2017

General Comments

This paper is a model comparison of 10 ESMs with satellite data from MODIS. The novel contribution of this paper is that is breaks down the analysis of model differences and biases by PFT. Similar comparisons of NBP, GPP, and CUE have been done already, as described in the Introduction section (Anav et al. 2013, Shao et al. 2013, Zhang et al. 2014).

I thought an interesting and underemphasized point was that reduced C uptake due to N limitation decreased CUE in models that added a representation of the N cycle, bringing up questions about whether we need coupled C and N or if so, whether the implementation in these models (reducing C uptake) is correct.

[Figure]

The writing is clear and the methodologies are described well, though the Results section is indeed largely results, as well as large portions of the Summary section. In the Specific Comments I asked some questions in places where I thought more discussion was needed. I think this paper could be improved with either a dedicated Discussion section or more development of the main findings and their implications. To me, the most interesting points in the paper are that: ESMs are biased in similar ways, CUE depends on climate (and therefore indirectly on PFT?), and the bias pattern of CUE differs in C-N coupled models.

Specific Comments

L63-68 That MME-values of CUE are dependent on temperature but MODIS-derived CUE is not suggests to me that we don't know what controls CUE in the real world. This is perhaps the more intriguing call to action than better model validation (because these models are clearly deficient, the question is why?).

L238 Clarify that "the only exception" refers to NPP calculated by GPP-Ra, not to the previous sentence about nitrogen cycling.

L279-293 Just to clarify: Did you use the model runs with their native PFTs, but then make Figures 12, S4 by breaking down the output by the classifications in Figure 1?

L326 I like Taylor diagrams, but I think you need a citation for the method.

L354 What do you mean by this statement? I think you need to clarify what "higher model performance" would be in this case (e.g., greater precision or accuracy). I also think that you could have large biases in individual models but still get a good mean estimate as long as those biases are different and random in each model, but I think your point is that many of the models are biased in similar ways.

L378-388 It is hard to identify whether biases are due to parameterization, or the climate forcings. Are there papers that use identical climate forcings to diagnose biases? I love to see some discussion here that tries to diagnose where biases come from using

previous studies.

L395 This may be the motivation for why you used it here, but it not the only reason to care about CUE: it is also an important control on the C cycle and may change under future climate or with land use.

L402-403 Explanation for why CUE is higher in cold weather? Growth more limited by access to C and respiration more limited by temperature?

L444 How do you know that nitrogen limitation effects CUE more than temperature and precipitation?

L450-453 The explanation for high needleleaf CUE in this paragraph is just a definition of CUE, can you provide a biophysical explanation?

L464-470 I think it's very interesting that models with C and N cycles simulate lower CUE. Seems to agree with theory that N limitation lowers CUE (Sinsabaugh et al. 2013 Ecology Letters).

L495-496 By what mechanism do these plants increase CUE?

L500 Seems from Figure 14 that the opposite is true: that ESMs don't respond enough to temperature, especially at >20C.

L527-529 Here you say that the parameterization is more important than the climate, which seems to be in contradiction to the uncertainty in L378-388.

Figures A lot of 9 panel maps are hard to look at and not all seem necessary. For example, Figure 10 does a much better job of summarizing the main points of CUE spatial distribution than doe Figure 9 (which could probably go in the Supplement).

Technical Comments L451 Do you mean "deciduous forests" instead of "dense forests"

L540 "In different with MODIS", suggest edit to "In contrast with MODIS".

Figures Can you label the multi-panel figures with letters (i.e., a,b,c..)

Figure 5 Remove title.

Figure 6 Why is there a red MODIS symbol in the legend?

Figure S4 Change 1. and 2. in legend to blue triange and green circle, respectively.

———————————————————

---

## Referee Comment (RC2) · Anonymous Referee #2 · 2 Feb 2017

Summary: This paper compares the GPP, NPP, and CUE of 10 CMIP5 models to an 'observation' from MODIS, and evaluates their similarities and differences. Model precipitation and temperature fields are similar to reanalyses, but carbon cycle products are diverse.

At the outset, I have a major problem with this paper. The only observation MODIS makes is radiances recorded by reflected sunlight in multiple spectral bands. MODIS does not 'observe' LAI, it calculates it with a model. MODIS does not 'observe' fPAR, it calculates it with an NDVI-type algorithm. MODIS does not 'observe' GPP, it models it using a light-response model (fundamentally different than the enzyme-kinetic models from CMIP5 that are evaluated here). MODIS most certainly does not 'observe' any

form of respiration (Ra or Rh), but calculates these as a function of modeled GPP and equations that relate respiration dependence to temperature and moisture. To claim that MODIS-derived quantities are somehow correct compared to other models is just about impossible to defend. We know that MODIS has biases in that the radiances are masked during times of even fairly thin cloud optical depth. This has major impacts in the tropics (where area-mean GPP is just about the highest on the globe), in savanna regions where GPP can respond strongly and rapidly to seasonal rains, and surely in other regions as well. MODIS GPP/NPP/CUE is just another model. With this perspective, there's not a lot here other than demonstrating that the models are different, but we knew that already. Some of that analysis was already done in the Shao paper.

Furthermore, there are multiple instances where the grammar and prose are incorrect, especially in the conclusions section. I appreciate that to write a scientific manuscript in a non-native language can be a challenge, but there are resources to help with this. The authors must find and utilize these resources to ensure that the manuscript meets English language grammar and usage requirements. Peer-review is for scientific content, not proofreading.

For these reasons, I must recommend rejection of this manuscript. The assumption of MODIS 'correctness' permeates the document, and the grammar errors are too many to mention.

Suggestion: The authors have obviously put a lot of work into the analysis, and I believe it may have value as a resubmission. The idea of a comparison between the light-response MODIS model and the various enzyme-kinetic CMIP5 models interests me. I have not worked with the CMIP5 data directly myself, but I imagine that gridded $CO_2$ values are part of the output suite. I wonder if it would be possible to pull out gridcells that contain $CO_2$ flask sites and compare mean annual cycles from the models to those observations? I'm pretty sure I've seen MODIS-based predictions of global $CO_2$ fields, and this might provide an interesting comparison of models to an actual observation. I looked for an article like this in the currently published CMIP5

literature, but did not find one. Perhaps an opportunity exists here. I do not think a full inversion or data assimilation project would be necessary, but perhaps something like Wang et al. (2016), where comparisons were made between models and observed CO2 concentration at several flux sites.

Specific comments: • Map figures should be broken at the date line, not the prime meridian. Currently, a large fraction of the middle of these plots is blank Pacific Ocean, and it is hard to discern behavior in Africa and Europe. • Figure 2: it is not necessary to show both the 2000-2005 and 1983-2005 maps. You can show the 2000-2005 map and just say that the longer-period map is similar. • Reanalyses of temperature and humidity are observationally-based, but precipitation, even for a reanalysis is based at least to some extent on models. I'm not sure how reliable any global precipitation map is, even the observational/satellite products like GPCP or TRMM. 'Reanalysis' might be a better description than 'observation' in graphs like Figure 3.

Reference: Wang, Y., N.M. Deutscher, M. Palm, T. Warneke, J. Nothold, I. Baker, J. Berry, P. Suntharalingam, N. Jones, E. Mahieu, B. Lejeune, J. Hannigan, S. Conway, J. Mendonca, K. STrong, J.E. Campbell, A. Wolf, S. Kremser, 2016: Towards understanding the variability in biospheric CO2 fluxes: using FTIR spectrometry and a chemical transport model to investigate the sources and sinks of carbonyl sulfide and its link to CO22. Atmos. Chem. Phys., 16, 2123-2138, doi:10.5194/acp-16-2123-2016.
* * *

---

## Author Comment (AC1) · 11 Mar 2017

General Comments This paper is a model comparison of 10 ESMs with satellite data from MODIS. The novel contribution of this paper is that is breaks down the analysis of model differences and biases by PFT. Similar comparisons of NBP, GPP, and CUE have been done already, as described in the Introduction section (Anav et al. 2013, Shao et al. 2013, Zhang et al. 2014).

I thought an interesting and underemphasized point was that reduced C uptake due to N limitation decreased CUE in models that added a representation of the N cycle, bringing up questions about whether we need coupled C and N or if so, whether the implementation in these models (reducing C uptake) is correct.

The writing is clear and the methodologies are described well, though the Results section is indeed largely results, as well as large portions of the Summary section. In the Specific Comments, I asked some questions in places where I thought more discussion was needed. I think this paper could be improved with either a dedicated Discussion section or more development of the main findings and their implications. To me, the most interesting points in the paper are that: ESMs are biased in similar ways, CUE depends on climate (and therefore indirectly on PFT?), and the bias pattern of CUE differs in C-N coupled models.

Specific Comments L63-68 That MME-values of CUE are dependent on temperature but MODIS-derived CUE is not suggests to me that we don't know what controls CUE in the real world. This is perhaps the more intriguing call to action than better model validation (because these models are clearly deficient, the question is why?).

⇒ Actually, the sensitivity of CUE is not only function of temperature (Tucker et al., 2013) but also nitrogen availability (Zha et al., 2013). However, most existing ESMs don't consider the nitrogen cycle except CESM-BGC and NorESM. Moreover, ESMs adapted the nitrogen cycle are not perfect (e.g., nitrogen fluxes and amounts are too much dependent on carbon fluxes and amount in the models). This might lead to a non-linearity and complex relationship between CUE and temperature in the real case. In addition, the parameterization of terrestrial carbon cycle in ESMs is imperfect. For instance, most ESMs adjusted the vegetation growth by the minimum of carbon, water, light limitation based on Farquhar et al. (1980). We have included additional discussion in the revised manuscript (L591-599).

L238 Clarify that "the only exception" refers to NPP calculated by GPP-Ra, not to the previous sentence about nitrogen cycling.

⇒ This part might be the Line 268. Expect for MRI-ESM, NPP is calculated by GPP and Ra from dynamical parameterization method. In MRI-ESM, the terrestrial carbon cycle is calculated by empirical methods derived from precipitation and temperature.

This is major difference of simulated NPP from MRI-ESM with other ESMs. We clarify the sentence (L274)

L279-293 Just to clarify: Did you use the model runs with their native PFTs, but then make Figures 12, S4 by breaking down the output by the classifications in Figure 1?

⇒ We classified model PFTs using observed MODIS MCD12Q1 land cover classification data (L181-L189 and figure 1). We mentioned it with related issues in L285-286.

L326 I like Taylor diagrams, but I think you need a citation for the method.

⇒ We added the reference in Taylor diagram method (L336 and references page).

L354 What do you mean by this statement? I think you need to clarify what "higher model performance" would be in this case (e.g., greater precision or accuracy). I also think that you could have large biases in individual models but still get a good mean estimate as long as those biases are different and random in each model, but I think your point is that many of the models are biased in similar ways.

⇒ We clarify the meaning in the manuscript ( L359-L367). "Unlike the cases in temperature and precipitation, the pattern correlation of the MME in terms of GPP and NPP is not necessarily higher than that of the individual models in this case. This suggests the presence of similar type of systematic model deficiencies in current CMIP5 ESMs, which is even larger than random individual model errors supposed to be cancelled out through the multi-model ensemble average."

L378-388 It is hard to identify whether biases are due to parameterization, or the climate forcings. Are there papers that use identical climate forcings to diagnose biases? I love to see some discussion here that tries to diagnose where biases come from using previous studies.

⇒ This is a very interesting point suggested by the reviewer. The multi-model comparison could be possible by driving off-line LSM models with the same climate forcing. To our knowledge there has been no study related with this problem. Instead, we have

included some relevant discussion from Mao et al. (2010) in the revised manuscript as below (L391-L395): "Mao et al. (2010) showed a quite similar bias pattern in GPP from their offline CLM4 experiment with observed climate forcing to the pattern of CESM1-BGC shown in this study (e.g., positive over tropics and negative over northern hemisphere high latitudes). This implies that the uncertainty in climate forcing is not a primary one for the GPP biases"

L395 This may be the motivation for why you used it here, but it not the only reason to care about CUE: it is also an important control on the C cycle and may change under future climate or with land use.

⇒ Following your comment, we emphasize this aspect in the manuscript (L412-413).

L402-403 Explanation for why CUE is higher in cold weather? Growth more limited by access to C and respiration more limited by temperature?

⇒ Ise et al. (2010) and Bradford and Crowther (2013) suggested that CUE could be limited substantially by overly-sensitive autotropic respiration by plants in warm climate based on their observational studies. We added this in the text (L416-419).

L444 How do you know that nitrogen limitation effects CUE more than temperature and precipitation?

⇒ This study suspect that the nitrogen limitation is affecting the simulated CUE more in the model rather than temperature or precipitation. Figure 13 shows the uniform increase of CUE in the Only C experiment. Moreover, Fig. S4 shows that this increase tends to occur in all PFTs. The temperature and precipitation change between the two runs are regional and they are not able to explain this globally-uniform signal, such as in southern hemisphere (See the supplementary figure 1 below).

L450-453 The explanation for high needleleaf CUE in this paragraph is just a definition of CUE, can you provide a biophysical explanation?

⇒ We added biophysical explanation of needleaf forest (L472), which is defined as

gymnosperms.

L464-470 I think it's very interesting that models with C and N cycles simulate lowerCUE. Seems to agree with theory that N limitation lowers CUE (Sinsabaugh et al. 2013Ecology Letters).

⇒ Yes. I agree with your opinion. Pervasive nutrient (e.g., Nitrogen) may induce the lower CUE values in this study.

L495-496 By what mechanism do these plants increase CUE?

⇒ We clarify the sentence (L517-518). Vegetation in cold climate with less rainfall is more efficient in storing carbon for growing and maintenance, which tends to increase CUE. Vegetation in warmer climate is overly sensitive to temperature in respiration (Ise et al. 2010); Bradford and Crowther 2013), which tends to lower CUE.

L500 Seems from Figure 14 that the opposite is true: that ESMs don't respond enough to temperature, especially at >20C.

⇒ Red dots (MME) is more sensitive to temperature, particularly in EBF and GRA over 20 C, which encompass largest area.

L527-529 Here you say that the parameterization is more important than the climate, which seems to be in contradiction to the uncertainty in L378-388.

⇒ As we mentioned in the above, the systematic biases in the models may reflect the uncertainties in the parameterized carbon cycles, as well as in the simulated climates. However, this study emphasizes more important role of parameterization in explaining the CMIP5 model diversity in GPP. Indeed, Mao et al. (2010)'s GPP pattern driven from observed climate forcing does not make significant difference from fully interactive simulations that might drift the climate from the observation. This implies that the uncertainty in climate forcing is not a primary one for the GPP biases

Figures A lot of 9 panel maps are hard to look at and not all seem necessary. For

example, Figure 10 does a much better job of summarizing the main points of CUE spatial distribution than does Figure 9 (which could probably go in the Supplement).

⇒ Although we agreed with the reviewers comment, the display of 9-panel maps is an inevitable choice to address the model differences in regional scale. This is basically driven by the spatial variation of PFTs, which is our major theme of this study.

Technical Comments

L451 Do you mean "deciduous forests" instead of "dense forests" L540 "In different with MODIS", suggest edit to "In contrast with MODIS".

⇒ We modified L546

Figures Can you label the multi-panel figures with letters (i.e., a,b,c..)

⇒ We added labels in figures.

Figure 5 Remove title.

⇒ We removed title.

Figure 6 Why is there a red MODIS symbol in the legend?

⇒ We removed MODIS symbol

Figure S4 Change 1. and 2. in legend to blue triange and green circle, respectively.

⇒ We changed these symbols.

References:

Amthor, J. S.: The McCree–de Wit–Penning de Vries–Thornley respiration paradigms: 30 years later, Annals of Botany, 86, 1–20, 2000. Bradford, M. A., Keiser, A. D., Davies, C. A., Mersman,n C. A., Strickland, M. S.: Empirical evidence that soil carbon formation from plant inputs is positively related to microbial growth, Biogeochem., 113, 271–281, 2013. Farquhar, G. D., von Caemmerer, S., and Berry, J. A.: A biochemical

model of photosynthetic CO2 assimilation in leaves of C3 species, Planta, 149, 78–90, doi:10.1007/BF00386231, 1980. Ise, T., Litton, C. M., Giardina, C. P., Ito, A.: Comparison of modeling approaches for carbon partitioning: impact on estimates of global net primary production and equilibrium biomass of woody vegetation from MODIS GPP, J. Geophys, Res., 115: G040205, 2010. Mao, J. Thornton, P. E. and Shi, X.: Remote Sensing Evaluation of CLM4 GPP for the Period 2000–09, J. Clim., 25, 5327-5342, 2012. Tucker, C. L., Bell, J., Pendall, E., Ogle, K.: Does declining carbon-use efficiency explain thermal acclimation of soil respiration with warming?, Glob. Change Biol., 19, 252–263, 2013. Zha, T. S., Barr, A. G., Bernier, P. -Y., Lavigne, M. B., Trofymow. J. A., Amiro. B. D., Arain,M. A., Bhatti, J. S., Black, T. A., Margolis, H. A. McCaughey, J. H., Xing, Z. S., VanRees, K. C. J., Coursolle. C.: Gross and aboveground net primary production at Canadian forest carbon flux sites, Agricul. and Fore. Meteorol., 174–175: 54–64, 2013.

Please also note the supplement to this comment:
http://www.biogeosciences-discuss.net/bg-2016-536/bg-2016-536-AC1-supplement.pdf

a) CN-OnlyC

b) CN-OnlyC

**Fig. 1.** The temperature (top) and precipitation (bottom) differences between CN and Only C experiments using NCAR CESM.

---

## Author Comment (AC2) · 11 Mar 2017

Summary: This paper compares the GPP, NPP, and CUE of 10 CMIP5 models to an 'observation' from MODIS, and evaluates their similarities and differences. Model precipitation and temperature fields are similar to reanalyses, but carbon cycle products are diverse. At the outset, I have a major problem with this paper. The only observation MODIS makes is radiances recorded by reflected sunlight in multiple spectral bands. MODIS does not 'observe' LAI, it calculates it with a model. MODIS does not 'observe' fPAR, it calculates it with an NDVI-type algorithm. MODIS does not 'observe' GPP, it models it using a light-response model (fundamentally different than the enzyme-kinetic models from CMIP5 that are evaluated here). MODIS most certainly does not

'observe' any form of respiration (Ra or Rh), but calculates these as a function of modeled GPP and equations that relate respiration dependence to temperature and moisture. To claim that MODIS-derived quantities are somehow correct compared to other models is just about impossible to defend. We know that MODIS has biases in that the radiances are masked during times of even fairly thin cloud optical depth. This has major impacts in the tropics (where area-mean GPP is just about the highest on the globe), in savanna regions where GPP can respond strongly and rapidly to seasonal rains, and surely in other regions as well. MODIS GPP/NPP/CUE is just another model. With this perspective, there's not a lot here other than demonstrating that the models are different, but we knew that already. Some of that analysis was already done in the Shao paper.

⇒ We accept the MODIS is not the actual observation but another modeled estimated from satellite radiances. However, there are also many studies in the following suggesting that the gridded MODIS GPP is consistent well with in-situ data and good for the model validation. Moreover, for analyzing global distribution of net primary production (NPP), we have only one available option for the data, which is MODIS. In the original manuscript, we already compared global averaged GPP amount between FLUXNET-MTE and MODIS, showing very small difference less than 1 % of total mean value as well as consistent spatial variations.

- Heinsch et al. (2006) suggested that the tower-based observation using AmeriFlux and MODIS GPP reasonably compared for most vegetation types (L204-206). They also compared the seasonal cycle of GPP over all vegetation types. It also captures the rapid onset of leaf-on and out reasonably.

- Turner et al. (2006) compared MODIS GPP and in-situ observations over multiple years at three sites (e.g., boreal conifer forest, temperate deciduous forest and grassland). Interannual variation of GPP in MODIS agreed with that from ground-based observations.

- Zhao et al. (2005) also suggested that the MODIS GPP fits well with GPP from 12 flux towers over North America. Moreover, Zhao et al. (2006) compared with MODIS GPP forced by original DAO data, ERA-40 and NCEP reanalysis data and forced by observed weather station data in U. S. (n=321). The described GPP by DAO shows the best results compared with other reanalysis data (corr=0.94).

- Chen et al. (2014) evaluated MODIS GPP with different adjustment of parameters in algorithm compared with 21 tower measurements classifying 9 PFTs. Despite parameter change, MODIS GPP is able to capture R-squared values comparing with in-situ data (0.46~0.89).

- Verma et al. (2014) validated the MODIS GPP and FLUXNET data sets. Except croplands, the MODIS GPP reasonably captured spatial variation in annual mean GPP in every biome.

⇒ To warrant the use of MODIS for the model validation, we also compared carbon use efficiency (CUE) from our studies and previous studies using site-based observations (supplementary figure 1). DNF has the highest CUE values in our study, being consistent with all previous studies. In addition, the plants with short canopy height (SHR, GRA and CROP) have the values around 0.5 and needleleaf forest (ENF, DNF) shows relatively higher values than that of other PFTs, which are also consistent with the findings in other studies. This confirms that MODIS does not have any known significant defect to reject the model validation. We added this discussion L554-564, with the figure 1 in the revised manuscript (It is Table S1 in manuscript)

⇒ Moreover, we also evaluated the MODIS gridded GPP data and GPP station data from FLUXNET(http://fluxnet.fluxdata.org). The comparison of the GPP data at the 53 tower sites (See Supplementary figure 2 below) and same gridded area from MODIS GPP 1 x 1 degree resolution is shown in the supplementary figure 3 below. The r-squared value is about 0.56. It is comparable with other previous studies about evaluating MODIS satellite data.

[Figure]

⇒ Moreover, we compared GPP from CMIP5 ESMs at same sites in Supplementary figure 4 below. The range of r-squared values in CMIP5-ESMs is 0.25 – 0.43. It is significantly lower than MODIS satellite data. It means that the GPP from MODIS is comparable dataset to evaluate numerical models.

Furthermore, there are multiple instances where the grammar and prose are incorrect. I appreciate that to write a scientific manuscript in a non-native language can be a challenge, but there are resources to help with this. The authors must find and utilize these resources to ensure that the manuscript meets English language grammar and usage requirements. Peer-review is for scientific content, not proofreading. For these reasons, I must recommend rejection of this manuscript. The assumption of MODIS 'correctness' permeates the document, and the grammar errors are too many to mention.

⇒ We carefully revised the manuscript once again.

Suggestion: The authors have obviously put a lot of work into the analysis, and I believe it may have value as a resubmission. The idea of a comparison between the light-response MODIS model and the various enzyme-kinetic CMIP5 models interests me. I have not worked with the CMIP5 data directly myself, but I imagine that gridded $CO_2$ values are part of the output suite. I wonder if it would be possible to pull out gridcells that contain $CO_2$ flask sites and compare mean annual cycles from the models to those observations? I'm pretty sure I've seen MODIS-based predictions of global $CO_2$ fields, and this might provide an interesting comparison of models to an actual observation. I looked for an article like this in the currently published CMIP5 literature, but did not find one. Perhaps an opportunity exists here. I do not think a full inversion or data assimilation project would be necessary, but perhaps something like Wang et al. (2016), where comparisons were made between models and observed $CO_2$ concentration at several flux sites.

⇒ We appreciate your suggestions. However, the MODIS-based global $CO_2$ data are also used MODIS GPP and NPP data for input forcing to predict global CO2 data (Potter et al., 2012; Guo et al., 2012). It induces more complexity and uncertainty of carbon fluxes using MODIS satellite data. Moreover, the atmospheric CO2 concentration is closely related with carbon fluxes from terrestrial biosphere. Despite of not perfect, the evaluation of performance of simulation skill of terrestrial carbon cycle in CMIP5-ESMs is needed and valid to improve the carbon cycle in the numerical models and statement of our knowledge of terrestrial carbon cycle.

⇒ Even though, MODIS GPP and NPP are light use based "model". It is best and only one data to evaluate global distribution of CUE in ESMs. For more accurate and realistic validation of numerical models, more fine spatial and temporal in-situ based observation data are needed. We added this discussion in L548-550 and L592.

Specific comments: Map figures should be broken at the date line, not the prime meridian. Currently, a large fraction of the middle of these plots is blank Pacific Ocean, and it is hard to discern behavior in Africa and Europe.

⇒ We modified it.

Figure 2: it is not necessary to show both the 2000-2005 and 1983-2005 maps. You can show the 2000-2005 map and just say that the longer-period map is similar.

⇒ We decided to keep the figure as in the original, as the data sampling issue might be an important issue to some of readers concerning the interannual variability such as ENSO. Reanalyses of temperature and humidity are observationally-based, but precipitation, even for a reanalysis is based at least to some extent on models. I'm not sure how reliable any global precipitation map is, even the observational/satellite products like GPCP or TRMM

⇒ The surface air temperature and precipitation data are not based on model or reanalysis but the gridded observations from CRU (L241-245).

| | Kim et al. (2017) | Delucia et al. (2007) | Amthor (2000) | Choudhury (2000) | Zhang et al. (2009) | Average (STD) |
|---|---|---|---|---|---|---|
| ENF | 0.59 | 0.41 | 0.61 | - | 0.56 | 0.54 (0.09) |
| EBF | 0.41 | 0.32 | 0.54 | 0.42 | 0.32 | 0.40 (0.09) |
| DNF | 0.63 | 0.59 | 0.76 | - | 0.59 | 0.64 (0.08) |
| DBF | 0.42 | 0.46 | 0.67 | - | 0.51 | 0.52 (0.11) |
| MF | 0.60 | 0.45 | - | - | 0.41 | 0.49 (0.10) |
| SHR | 0.54 | - | 0.50 | 0.45 | 0.52 | 0.50 (0.04) |
| GRA | 0.54 | - | 0.49 | 0.52 | 0.51 | 0.51 (0.02) |
| CROP | 0.52 | - | 0.45 | 0.56 | 0.52 | 0.51 (0.05) |

**Fig. 1.** Comparison of averaged CUE for each PFTs. ENF is evergreen needleleaf forest, EBF is evergreen broadleaf forest, DNF is deciduous needleleaf forest, DBF is deciduous broadleaf, MF is mixed forest, SHR

| STN | Lon | Lat | STN | Lon | Lat | STN | Lon | Lat | STN | Lon | Lat |
|---|---|---|---|---|---|---|---|---|---|---|---|
| BE-Bra | 4.52 | 51.31 | DE-Hai | 10.45 | 51.08 | IT-SRo | 10.28 | 43.73 | US-Syv | -89.35 | 46.24 |
| BE-Vie | 6.00 | 50.31 | DE-Geb | 51.10 | 10.91 | NL-Loo | 5.74 | 52.17 | US-Ton | -120.9 | 38.43 |
| BR-Sa3 | -54.97 | -3.02 | DE-Tha | 13.57 | 50.96 | RU-Fyo | 32.92 | 56.46 | US-UMB | -84.71 | 45.56 |
| CA-NS1 | -98.48 | 55.88 | DK-Sor | 11.64 | 55.49 | US-Blo | -120.6 | 38.90 | US-Var | -120.9 | 38.41 |
| CA-NS2 | -98.52 | 55.91 | DK-ZaH | -20.55 | 74.47 | US-Cop | -109.3 | 38.09 | US-WCr | -90.08 | 45.81 |
| CA-NS3 | -98.38 | 55.91 | FI-Hyy | 24.30 | 61.85 | US-GBT | -106.2 | 41.37 | US-Wi0 | -91.08 | 46.62 |
| CA-NS4 | -98.38 | 55.91 | FI-Jok | 23.51 | 60.90 | US-Ha1 | -72.17 | 42.54 | US-Wi3 | -91.10 | 46.63 |
| CA-NS5 | -98.49 | 55.86 | FI-Sod | 26.64 | 67.36 | US-Los | -89.98 | 46.08 | US-Wi4 | -91.17 | 46.74 |
| CA-NS6 | -98.96 | 55.92 | FR-Pue | 3.60 | 43.74 | US-MMS | -86.41 | 39.32 | US-Wi6 | -91.30 | 46.62 |
| CA-NS7 | -99.95 | 56.64 | IT-Col | 13.59 | 41.85 | US-Ne1 | -96.48 | 41.17 | ZA-Kru | 31.50 | -25.02 |
| CA-Qfo | -74.34 | 49.69 | IT-Cpz | 12.38 | 41.71 | US-Ne2 | -96.47 | 41.16 | ZM-Mon | 23.25 | -15.44 |
| CA-SF2 | -105.8 | 54.25 | IT-La2 | 11.29 | 45.95 | US-Ne3 | -96.44 | 41.18 | | | |
| CA-SF3 | -106.0 | 54.09 | IT-Ren | 11.43 | 46.59 | US-NR1 | -105.5 | 40.03 | | | |
| CH-Dav | 9.86 | 46.82 | IT-Ro1 | 11.93 | 42.41 | US-PFa | -90.27 | 45.95 | | | |

**Fig. 2.** Locations of 53 FLUXNET GPP tower sites.

GPP (gCm²/d)

Y = 0.44x + 419.88
R² = 0.56

**Fig. 3.** Comparison of averaged GPP measured for 6 years (2000-2005) at the 53 tower sites and MODIS GPP gridded areas which are coincided with tower sites.

[Figure]

Legend:
- MODIS
- MME
- BCC-CSM1
- BCC-CSM1M
- CanESM2
- CESM1-BGC
- ESM2G
- ESM2M
- MIROC-ESM
- MPI-ESM-LR
- MRI-ESM1
- NorESM1-ME

**Fig. 4.** R-squared values of GPP at 53 tower sites, MODIS and 10 CMIP5-ESMs.

**Supplement:**

[revised manuscript text omitted]

However, these are the best and only available data for the validation of global distribution of CUE simulated by ESMs.

In Table S1, we compared

CUE from our studies and previous studies that used the site- based observation data . DNF shows highest CUE values in this study, which is consistent well with the findings in previous studies. In addition, the plants with short canopy height (SHR, GRA and CROP) show the valuethe needleleaf forests (ENF, DNF)

show the values relatively higher than those of other PFTs consistently.

Analyzing

CUE help us to understand the carbon storage in simulated terrestrial ecosystem in ESMs. At first, the spatial distribution of observed CUE from space (e.g., MODIS) depends on climate condition such as precipitation and temperature. For example, the regions of large precipitation and warm climate show low CUE, while the regions of dry and cold climate show high CUE.

It indicates that CUE at the regions with warm temperature and abundant precipitation could be lowered as there is a plenty of production and plant growth. The vegetation in cold temperature and insufficient precipitation adapts to the environmental condition for survival by increasing CUE.

  In  contrast with MODIS, we found clear difference of CUE between ESMs. The bias pattern of two ESMs from BCC showed the hemispheric contrast to positive in NH and negative in SH. The strong negative bias of CUE over southern hemisphere is shown in GFDL's models. The CUE in ESMs based on CLM4 (e.g., CESM-BGC and NorESM-ME) are is significantly underestimated globally. This large uncertainty of CUE in individual models is influenced by biogeochemical parameterization of land surface model. In the MME, the spatial distribution of CUE is reasonably simulated. However, sStrong negative bias is found over

Amazon, which is . It is caused by that unbalanced ratio of GPP and Ra in the terrestrial carbon fluxes over tropical forest such as evergreen broadleaf forest in the most models. The inverse relationship between temperature and CUE is reasonably simulated in the MME over dry regions. Generally, Ra is more sensitive to temperature than GPP in the real world over a certain range of temperatures (Woodwell et al., 1990; Ryan, 1991; Piao et al., 2010). It means suggests that the sensitivity of temperature to photosynthesis is weaker than that of respiration (Arnone and Korner, 1997; Enquist et al., 2007).    Actually, the sensitivity of CUE is not only a function of temperature (Tucker et al., 2013) but also a function of nitrogen availability (Zha et al.,

2013). This might lead to a non-linearity and complex relationship between CUE and temperature in the real case. However, most ESMs in CMIP5 do not consider the nitrogen cycle except CESM-BGC and NorESM. Most existing ESMs tend to adjust the vegetation growth by the minimum of carbon, water, light limitation based on Farquhar et al. (1980). Moreover,

ESMs adapted the nitrogen cycle are not perfect in their parameterizations. For instance, nitrogen fluxes and amounts are too much dependent on carbon fluxes and amount in the models.

[revised manuscript text omitted]

3, 255–270, 1968.

Leith, C. E.: Climate response and fluctuation dissipation, J. Atmos. Sci., 32, 2022-2026, doi:10.1175/1520-0469(1975)032<2022:CRAFD>2.0.CO;2, 1975.

Mao, J. Thornton, P. E. and Shi, X.: Remote Sensing Evaluation of CLM4 GPP for the Period

2000–09, J. Clim., 25, 5327-5342, 2012.

Monteith, J.: Solar radiation and productivity in tropical ecosystems, J. Appl. Ecol., 9, 747-

766, doi:10.2307/2401901, 1972.

Obata, A.: Climate carbon cycle model response to freshwater discharge into the North

Atlantic, J. Clim., 20, 5962–5976, doi:10.1175/2007JCLI1808.1, 2007.

Piao, S., Luyssaert, S., Ciais, P., Janssens, I. A., Chen, A., Cao, C., Fang, J., Friedlingstein,

P., Luo, Y., and Wang, S.: Forest annual carbon cost: a global-scale analysis of autotrophic
respiration, Ecol., 91, 652–661, doi:10.1890/08-2176.1, 2010.

Rahman, A. F., Sims, D. A., Cordova, V. D., and El-Masri, B. Z.: Potential of MODIS EVI
and surface temperature for directly estimating per-pixel ecosystem C fluxes, Geophys. Res.
Lett., 32, L19404, doi:10.1029/2005GL024127, 2005.

Running, S.W. and Gower, S.T.: FOREST-BGC, a general model of forest ecosystem
processes for regional applications. II. Dynamic carbon allocation and nitrogen budgets, Tree
Physiol., 9, 147–160, doi:10.1093/treephys/9.1-2.14, 1991.

Ryan, M. G.: Effects of climate change on plant respiration. Ecol. Appl., 1, 157–167, doi:
10.2307/1941808, 1991.

Shao P., Zeng, X., Sakaguchi, K., Monson, R. K., and Zeng, X.: Terrestrial carbon cycle:
climate relations in eight CMIP5 earth system models, J. Clim., 26, 8744-8764,
doi:10.1175/JCLI-D-12-00831.1, 2013.

Taylor, K. E.: Summarizing multiple aspects of model performance in a single diagram, J.
Geophy. Res., 106, 7183-7192, doi: 10.1029/2000JD900719, 2001.

Taylor, K. E., Stouffer, R. J., and Meehl, G. A.: An overview of CMIP5 and the experiment
design, Bull. Amer. Meteor. Soc., 93, 485–498, doi:0.1175/BAMS-D-11-00094.1, 2012.

Tucker, C. L., Bell, J., Pendall, E., Ogle, K.: Does declining carbon-use efficiency explain
thermal acclimation of soil respiration with warming?, Glob. Change Biol., 19, 252–263,
2013.

Turner, P. D., Ritts, W. D., Zhao, M., Kurc, S. A., Dunn, A. L., Wofsy, S. C., Small, E. E.,
and Running, S. W.: Assessing Interannual Variation in MODIS-Based Estimates of Gross
Primary Production, IEEE Trans. Geosci. Remote Sens., 44, 1899-1907, 2006.

[revised manuscript text omitted]

CROP (0.73).